**Investigation**

# The recombination landscape of the barn owl, from families to populations

Alexandros Topaloudis [ID],[1,2,*] Tristan Cumer [ID],[1,2] Eléonore Lavanchy [ID],[1,2] Anne-Lyse Ducrest [ID],[1] Celine Simon,[1] Ana Paula Machado [ID],[1] Nika Paposhvili [ID],[3] Alexandre Roulin [ID],[1] Jérôme Goudet [ID] [1,2,*]

[1]Department of Ecology and Evolution, University of Lausanne, Lausanne 1015, Switzerland
[2]Swiss Institute of Bioinformatics, Lausanne 1015, Switzerland
[3]Institute of Ecology, Ilia State University, Tbilisi 0162, Georgia

*Corresponding author: Department of Ecology and Evolution, University of Lausanne, Lausanne, Switzerland. Email: alexandros.topaloudis@unil.ch;
*Corresponding author: Department of Ecology and Evolution, University of Lausanne, Lausanne Biophore, 1015 Lausanne, Switzerland. Email: jerome.goudet@unil.ch

Homologous recombination is a meiotic process that generates diversity along the genome and interacts with all evolutionary forces. Despite its importance, studies of recombination landscapes are lacking due to methodological limitations and limited data. Frequently used approaches include linkage mapping based on familial data that provides sex-specific broad-scale estimates of realized recombination and inferences based on population linkage disequilibrium that reveal a more fine-scale resolution of the recombination landscape, albeit dependent on the effective population size and the selective forces acting on the population. In this study, we use a combination of these 2 methods to elucidate the recombination landscape for the Afro-European barn owl (*Tyto alba*). We find subtle differences in crossover placement between sexes that lead to differential effective shuffling of alleles. Linkage disequilibrium-based estimates of recombination are concordant with family-based estimates and identify large variation in recombination rates within and among linkage groups. Larger chromosomes show variation in recombination rates, while smaller chromosomes have a universally high rate that shapes the diversity landscape. We find that recombination rates are correlated with gene content, genetic diversity, and GC content. We find no conclusive differences in the recombination landscapes between populations. Overall, this comprehensive analysis enhances our understanding of recombination dynamics, genomic architecture, and sex-specific variation in the barn owl, contributing valuable insights to the broader field of avian genomics.

Keywords: linkage map; linkage disequilibrium; meiotic recombination; heterochiasmy; hotspots

## Introduction

Homologous meiotic recombination (hereafter recombination) is the reciprocal exchange of genetic material between homologous chromosomes during the first meiotic division. The physical exchange, called a crossover (CO), generates the necessary tension between homologous chromosomes to ensure their proper segregation in the daughter cells. In most species, in the absence of a CO, the daughter cells have an aberrant number of chromosome pairs (aneuploidy) leading to deleterious effects (Koehler *et al.* 1996; Hassold and Hunt 2001; Hassold *et al.* 2007; Zickler and Kleckner 2015; Zelkowski *et al.* 2019). Hence, at least 1 CO is expected per chromosome (or chromosome arm), termed an "obligate CO."

Beyond contributing to the integrity of functional meiotic division, recombination has evolutionary consequences. It shuffles alleles between haplotypes, affecting the genomic composition of a population, which, in turn, can incur evolutionary benefits, including faster adaptation to a changing environment and more efficient selection (Hill and Robertson 1966; Otto and Lenormand 2002). However, recombination can increase the rate of mutations and chromosomal rearrangements or impede adaptation by breaking up beneficial combinations of alleles (Barton and

Charlesworth 1998; Arbeithuber *et al.* 2015; Halldorsson *et al.* 2019). In fact, it can often be beneficial to link haplotypes together by suppressing recombination between loci, for example during the formation of a sex chromosome (Wright *et al.* 2016; Charlesworth 2017) or through the maintenance of inversions capturing ecologically relevant phenotypes (Küpper *et al.* 2016; Rowan *et al.* 2019; Todesco *et al.* 2020). As a result, the evolutionary net outcome of recombination is context dependent and may differ between populations, sexes, and genomic coordinates (Stapley *et al.* 2017).

The variation in recombination rates along the genomic sequence affects almost all genome-wide processes. For instance, in some, but not all species, genetic diversity along the genome correlates with recombination rate. Multiple forces can explain this pattern such as linked selection, a mutagenic effect of recombination, and GC-biased gene conversion (gBGC) (Begun and Aquadro 1992; Duret and Galtier 2009; Flowers *et al.* 2012; Cutter and Payseur 2013; Campos *et al.* 2014; Halldorsson *et al.* 2019). Importantly, the strength of these forces can vary among species, and their relative contribution in shaping genetic diversity is still unclear. In addition, regions with high recombination rates are often enriched in foreign DNA from past introgressions both in plants and animals (Schumer *et al.* 2018; Martin *et al.* 2019;

Dreissig *et al.* 2020; Edelman and Mallet 2021). Beyond influencing these processes, variable recombination rates can confound inferences, like scans for selection based on genetic differentiation (Booker *et al.* 2020). Thus, in order to advance our understanding of the evolutionary consequences of recombination and distinguish it from other forces that vary along the genome, we need to quantify recombination and understand the sources of its variation in different species and under different evolutionary backgrounds.

However, quantifying recombination is a challenging task. One approach is linkage mapping, the positioning of markers along the sequence with a distance proportional to the recombination rate between them. This approach requires family data (or controlled crosses, when available) and uses the cosegregation of alleles from one generation to the next, to identify the contemporary COs in the sample. Linkage mapping has been applied in several species thus far, providing a reliable measure of CO frequency (Kong *et al.* 2002; Stapley *et al.* 2017; Brazier and Glémin 2022). Linkage mapping enables quantification of the differences in recombination rates between sexes (i.e. heterochiasmy; Kong *et al.* 2010; Johnston *et al.* 2017; Brekke *et al.* 2022). However, historically, linkage mapping studies used a small set of genomic markers (Stapley *et al.* 2017) and the resolution of the method is limited by the number of observed meioses, making it impossible to quantify recombination accurately at a fine scale with the sample sizes available in many nonmodel species (Halldorsson *et al.* 2019).

Another approach used to estimate recombination rates uses whole genome sequences or reduced representation genotyping from tens of unrelated individuals and models the observed linkage disequilibrium (LD) between markers (Auton and McVean 2007; Chan *et al.* 2012; Spence and Song 2019). This method (hereafter referred to as LD-based inference) assesses ancestral recombination events that occurred in the coalescent history of the samples (Li and Stephens 2003; Stumpf and McVean 2003) and has enabled the quantification of fine-scale recombination variation (at the kilobase scale). These fine-scale inferences have identified that in many species COs occur in "hotspots," defined as regions with elevated recombination rate scattered along the genome (Myers *et al.* 2005; Mézard 2006; Paape *et al.* 2012; Choi and Henderson 2015). This observation is true for most species studied to date but not all (e.g. *Caenorhabditis elegans* and *Drosophila*; Kaur and Rockman 2014; Smukowski Heil *et al.* 2015). While initial discoveries in humans and mice implicated the fast-evolving *PRDM9* gene for the localization of hotspots along the sequence through motif matching, most species, notably birds, amphibians, most arthropods, and plants, do not have a functioning copy of the gene (Baudat *et al.* 2010; Myers *et al.* 2010; Parvanov *et al.* 2010; Baker *et al.* 2017). In these organisms, hotspots are found in accessible chromatin, near transcription start sites (TSSs) and CpG islands (CGIs), and are usually conserved even after millions of years of species' divergence (Auton *et al.* 2013; Lam and Keeney 2015; Singhal *et al.* 2015).

Despite the ability of LD-based inference to quantify fine-scale variation using a small number of genomes, the method has certain limitations. It infers the population recombination rate (rho), the product of the effective population size ($N_e$) and the recombination rate, rather than estimating the recombination rate directly. This has 2 major implications for the use of LD-based inference: (1) it does not distinguish between crossing over in male and female meioses and (2) it is affected by forces that change $N_e$ and not the recombination rate itself. The consequences of the latter are that forces that modify $N_e$, such as selection and fluctuating population sizes, can confound estimates of recombination (O'Reilly *et al.* 2008). Even if recent methods account for demography (Spence and Song 2019), estimates of recombination based on LD often need to be validated with a different method, such as linkage mapping (McVean *et al.* 2004; Axelsson *et al.* 2012; Shanfelter *et al.* 2019; Wall *et al.* 2022). While a combination of approaches is the preferred route to accurately infer the recombination rates, this has rarely been used in practice beyond in model species.

Birds have been the target of multiple evolutionary studies on speciation, hybridization, phenotypic microevolution, and biogeography (e.g. Poelstra *et al.* 2014; Vijay *et al.* 2016; Bosse *et al.* 2017; Enbody *et al.* 2023) but we still know very little about recombination variation in this class. Most of the current information on recombination in birds is from linkage mapping and cytological studies, which show that recombination exhibits broad-scale among-species variation and inconclusive patterns of sex differences (Groenen *et al.* 2009; Backström *et al.* 2010; Kawakami *et al.* 2014; van Oers *et al.* 2014; Malinovskaya *et al.* 2018, 2020; Hagen *et al.* 2020; Peñalba *et al.* 2020; Robledo-Ruiz *et al.* 2022; McAuley *et al.* 2024; Tan *et al.* 2024). Rates of recombination inferred from linkage mapping tend to differ between species, despite a rather conserved avian karyotype (Ellegren 2010; Bravo *et al.* 2021; Waters *et al.* 2021). For example, 2 members of the passerine order [collared flycatcher (*Ficedula albicollis*) and the superb fairy-wren (*Malurus cyaneus*)] show a 2-fold difference in genetic length for their largest syntenic chromosome despite a similar physical length (Kawakami *et al.* 2014; Peñalba *et al.* 2020). Further, birds show no consistent patterns of sex differences in recombination with the direction and strength of heterochiasmy varying between species (Malinovskaya *et al.* 2020; Sardell and Kirkpatrick 2020; McAuley *et al.* 2024; Tan *et al.* 2024). However, until recently, conclusions on heterochiasmy were only based on the total recombination frequency (genetic length) summed over all chromosomes in each sex. Recent studies have found evidence for differences in the placement of COs between sexes suggesting that heterochiasmy could be more subtle (McAuley *et al.* 2024; Tan *et al.* 2024; Zhang *et al.* 2024). Considering that the reasons behind heterochiasmy are not well understood, and that multiple adaptive hypotheses have been proposed (Lenormand and Dutheil 2005; Mank 2009; Brandvain and Coop 2012; Sardell and Kirkpatrick 2020), the avian class provides fertile ground for research on the forces driving heterochiasmy. Finally, since all avian species studied so far—with the exception of chickens (*Gallus gallus*)—belong to the passerine order, the sampled diversity may not be truly representative of the complete avian class.

Here, using both linkage mapping and LD information, we present the first recombination landscape for a species of the Strigiformes order, the barn owl (*Tyto alba*). This species has a high-quality genome assembly (Ducrest *et al.* 2020; Machado, Cumer, *et al.* 2022), a set of whole genome sequences available from previous studies (Cumer, Machado, Dumont, *et al.* 2022; Cumer, Machado, Siverio, *et al.* 2022; Machado, Cumer, *et al.* 2022; Machado, Topaloudis, *et al.* 2022; Cumer *et al.* 2024), and a long-term pedigreed population (Roulin 1999) with an untapped genomic potential (Charmantier *et al.* 2014; Sheldon *et al.* 2022). We capitalize on 176 genomes previously published, along with 326 newly sequenced individuals to build a high-confidence variant set of polymorphisms spanning the diversity of the species across the Western Palearctic. For recombination inference, we use linkage mapping on a subset of our data set, 250 owls belonging to 28 families, to (1) identify linkage groups (LGs) in the barn owl sequence assembly, (2) estimate the sex-averaged linkage map length, and (3) quantify sex differences in recombination.

Additionally, we employ an LD-based approach on 102 unrelated individuals from 3 populations to infer fine-scale recombination rate variation and scale our results using the estimates from the linkage map. With these complementary resources, we quantify variation in recombination between sexes as well as identify substantial differences in fine-scale patterns among chromosomes and populations.

## Methods

### Samples and sequencing

A total of 333 barn owl samples from Switzerland were sequenced for this study. In Western Switzerland, a subset of breeding barn owls has been monitored for over 30 years. During the breeding season, installed nest boxes are controlled for occupancy every 4 weeks. Individuals are ringed and measured, and a blood sample is taken to sex and genotype each individual (Py *et al.* 2006; Roulin *et al.* 2007; Antoniazza *et al.* 2010). Adult parents are also captured and sampled when possible. Given that barn owls rarely show extrapair paternity (Roulin *et al.* 2004), the unique ring identifiers of parents and offspring can be used to construct an observational pedigree.

Based on pedigree information, 285 individuals belonging to families between 1994 and 2020 were sequenced in 2020 and 2021. Families with more than 4 offspring and grandparent information were prioritized whenever possible. Sample DNA was extracted from blood using DNeasy Blood & Tissue Kit (QIAGEN) following manufacturer's instructions, quantified with dsDNA HS Qubit Kit (Thermo Fisher), and diluted to 6.3 ng/μL with 10 mM Tris-HCl, pH 8.0, in 40 μL. Libraries were prepared with Nextera DNA Flex (Illumina) and sequenced in Illumina HiSeq 4000 at the Lausanne Genomic Technologies Facility (GTF) giving an average depth of 12X (8–37X). This initial data set was increased using 6 samples from Georgia and 48 from Switzerland. The additional samples from Switzerland are from the same long-term study population, chosen so that they had the maximum number of descendants based on the field pedigree. These samples were sequenced in 2021. Sample preparation was as above, and sequencing was performed using Illumina NovaSeq 6000 with a produced depth of 30X (22–40X). All sequencing took place at the Lausanne GTF (University of Lausanne, Switzerland).

### Variant discovery and filtering

All available barn owl sequences were used for variant discovery. This included individuals mentioned above and samples from previous sequencing efforts (Cumer, Machado, Dumont, *et al.* 2022; Cumer, Machado, Siverio, *et al.* 2022; Machado, Cumer, *et al.* 2022; Machado, Topaloudis, *et al.* 2022; Cumer *et al.* 2024) along with 6 samples from Georgia and 3 from the island of Corsica (Supplementary File 2). In total, 502 samples were processed through the variant discovery pipeline described below. Raw reads were processed with *Trimmomatic* v0.39 (Bolger *et al.* 2014). Sequence adapters were removed, and reads with a length less than 70 bp were excluded. Mapping was performed with *BWA-MEM* v0.7.17 (Li 2013) on the barn owl genome assembly v.4.0 (https://www.ncbi.nlm.nih.gov/nuccore/JAEUGV000000000) (Machado, Cumer, *et al.* 2022), and read groups were added with *samtools* v1.15.1 (Li *et al.* 2009). Since the *GATK* v4.2.6 (Auwera *et al.* 2013) pipeline was used for variant discovery, base quality score recalibration was performed using a previously published variant "truth set" (Cumer, Machado, Dumont, *et al.* 2022). *GATK's Haplotype caller* was run with default parameters for each individual separately to generate individual *gvcf* files. These files

were merged, and joint calling was performed with all individuals together using *GenotypeGVCFs*. We initially identified 30,620,917 variants in the data set. Filtering focused on biallelic SNPs and consisted of the core technical filters suggested in the *GATK* pipeline, a "mappability" mask and a manual individual depth filtering. Specifically, technical filters included the following criteria: QD < 2.0, QUAL < 30, SOR > 3.0, FS > 60.0, MQ < 40.0, MQRankSum < −12.5, and ReadPosRankSum < −8.0. A further filtering was the exclusion of regions of the genome where our ability to confidently map reads is limited (i.e. a "mappability" mask) (Corval *et al.* 2023). Briefly, the reference genome was split into reads of 150 base pairs (bp) with a sliding of 1 bp. These artificial reads were mapped back to the reference using *BWA-MEM* v0.7.17. Regions of the reference sequence where less than 90% of the reads mapped perfectly and uniquely were discarded. This step masked 118 Mb of sequence that corresponds to 10% of the assembly, half of which belongs to scaffolds that did not make it into the linkage map, probably representing misassembled repeats. Variants were also filtered based on individual depth. A minimum and a maximum cutoff were applied. For the minimum cutoff, any genotype with less than 5 reads supporting it was set to missing (Benjelloun *et al.* 2019). For the maximum, a distribution of autosomal read depth per individual was extracted for a sample region (Super-Scaffold_1 and Super-Scaffold_2) with a length of 133.5 Mb. The mean and SD of depth was estimated, and any genotype with a read depth of more than 3 SD from the mean was set to missing to avoid the effect of repeated regions. After filtering, 26,933,469 variants were kept in 1,080 Mb of callable sequence, corresponding to 1 SNP per 40 bp. This is the raw variant data set used in all downstream analyses. Analysis-specific filters and number of SNPs used can be found below and in Supplementary Table 1 in Supplementary File 1.

### Pedigree and relatedness

The pedigree from observational data was confirmed with genomic information from a subset of the genome. SNPs from a sample subset of the genome, specifically 3 scaffolds (Super-Scaffold_11, 12, and 14), were filtered for minor allele count (>5) and missing data (<10%) and were pruned for LD using *plink* v.1.9 (Chang *et al.* 2015) with the command --*indep-pairwise* 100 10 0.1. This filtering created a data set with 91,874 SNPs. A genomic kinship matrix was calculated using the Weir and Goudet (2017) method as implemented in *hierfstat* v0.5-11 (Goudet 2005) R package. The kinship from genomic data was compared with the pedigree kinship, calculated using the *kinship2* v1.9.6 (Sinnwell *et al.* 2014) R package, and the pedigree was completed by manually fixing the first- and second-degree links when those could be resolved, for example identifying full siblings and parent–offspring links. Both k1 and k2 statistics, the probabilities of sharing 1 or 2 alleles, respectively, were calculated with the *SNPRelate* v.1.34.1 (Zheng *et al.* 2012) R library and were used to discern between relationships with the same kinship value (e.g. parent–offspring and full siblings). A set of unrelated individuals was selected automatically by pruning the genomic kinship table to only include individuals with a kinship of less than 0.03125. When selecting for unrelated individuals, we removed individuals with multiple high-kinship links and then we prioritized individuals with higher depth of coverage. This method left a subset of 187 unrelated individuals of which 76 were from the Swiss population.

### Linkage mapping

*Lep-MAP3* v0.2 (LM3) (Rastas 2017) was used to create a linkage map. A set of 250 individuals in 28 families was used, where a

family in LM3 is defined as a set of individuals around a unique mating pair. Most families had 4 offspring and 4 grandparents but numbers differed, ranging from 2 to 8 offspring and from 0 to 4 grandparents. To run LM3, a stringently filtered data set of biallelic SNPs was used. Specifically, we removed Mendelian incompatibilities using bcftools' *mendelian* plugin (Danecek *et al.* 2021) and retained a minimum of 5% MAF and a maximum of 5% missing data per site. We also filtered out SNPs that were less than 1,000 bp (1 kb) apart using *VCFtools* v.0.1.16 (Danecek *et al.* 2011). The first step of the LM3 pipeline *ParentCall* was used to transform the data into the appropriate LM3 format, and the options *halfSibs* and *removeNonInformative* were included. Data were filtered in LM3 using the *Filtering2* command, to shrink the size of the data set. Specifically, *dataTolerance* was set to 0.01 as suggested by the author, and *missingLimit* and *familyInformativeLimit* were set to 28. This meant that only variants that were nonmissing and informative in all families were kept. After filtering the data set, we retained 163,950 variants. We used *SeparateChromosomes* to identify the putative LGs based on a user-defined logarithm of odds (LOD) score cutoff. We selected a LOD score of 15 (for a justification see Supplementary Text 1 and Supplementary Fig. 1 in Supplementary File 1). Finally, *OrderMarkers* with the *usePhysical* option was executed. Ordering was repeated 3 times, and the output with the best Likelihood was selected for each LG. All 3 runs were compared to test for variation in estimated genetic maps.

After confirming that certain Super-Scaffolds correspond to the LGs, we used a larger set of variants to build the linkage map from. Specifically, we filtered on 10% missingness and 10% minor allele frequency, instead of the 5% filter used before. For all SNPs in each LG, we used *ParentCall* and *Filtering2* as before but ran *SeparateChromosomes* on each LG with a LOD cutoff of 5. Finally, we ran *OrderMarkers* by enforcing the physical map of the genome assembly and estimating the genetic map without de novo ordering by using the parameters *evaluateOrder* and *improveOrder*=0. This allowed us to use most variants on the scaffold (a total of 4,889,667 SNPs) and detect possible missing COs not identified by the previously filtered data set used in de novo ordering. A comparison of the 2 runs of *Lep-MAP3*, 1 with de novo marker ordering and 1 using the physical order, can be found in Supplementary Fig. 2 in Supplementary File 1.

In linkage mapping, certain markers might be erroneously mapped especially at the extremities of LGs. Thus, all markers with a separation larger than an arbitrary threshold of 2 cM in a region of 100 markers concentrated toward the ends of LGs were filtered out using a custom script adapted from *LepWrap* (Dimens 2022).

## LD recombination

To run *SMC++* v1.15.2 (Terhorst *et al.* 2017), we followed the authors' instructions as presented on the software's GitHub page (https://github.com/popgenmethods/smcpp). In summary, SNPs were filtered on 10% missingness, HWE (using a Fisher's exact test *P*-value cutoff of 0.05) and 5% minor allele frequency using *VCFtools* v.0.1.16. In addition, missing data were recoded using *plink* v1.9 (Chang *et al.* 2015), and the 5 samples with the highest coverage were selected as individuals to be provided to *SMC++*. The command *vcf2smc* was run for each of these 5 individuals. When executing "vcf2smc," the mappability mask was excluded by using the -m option. The model was estimated using all output files from the previous step and with a mutation rate of $1.93e^{-9}$ estimated from family data of a snowy owl *Bubo scandiacus* (Bergeron *et al.* 2023). The csv-formatted estimate of piecewise-constant $N_e$

in past generation intervals was used in subsequent *pyrho* (Spence and Song 2019) analyses.

We ran *pyrho* v0.1.7 with an unphased set of markers for 76 unrelated Swiss individuals. We used the data set of SNPs we used in *SMC++*, but we further filtered variants to be at least 10 bp apart using *VCFTools* v.0.1.16. The first step in the pyrho implementation was the precalculation of a 2-locus likelihood look-up table. This step takes into account the $N_e$ estimates from *SMC++* (Supplementary Fig. 3 in Supplementary File 1). For the Swiss samples, the number of diploid individuals was 76 and we used the Moran approximation parameter with a size of 200. The "hyperparameter" command was run to estimate metrics on the performance of different window sizes and block penalties. The authors' guidelines were followed on how to select the best combination of parameters. Briefly, we summed the Pearson correlation statistics outputted by pyrho and plotted their total sum against the L2 values. The authors suggest (https://github.com/popgenmethods/pyrho#hyperparam) that depending on the implementation, one might opt to choose the parameter combination that maximizes the correlation measures or minimizes L2. In our case, both conditions were satisfied with one combination of parameters and we run pyrho with that set of parameters. A table of the hyperparameter values for all populations can be found in Supplementary Table 2 in Supplementary File 1. With the inferred hyperparameters, the recombination rate was estimated using the "optimize" command on *vcfs* containing individual scaffolds that were previously filtered for singletons and a minimum distance of 10 bp between variants as in Wall *et al.* (2022).

## Downstream analyses

Putative centromeres were identified using a repeat annotation through a combination of 2 software, *RepeatObserver* v1 (Elphinstone *et al.* 2023) and *TRASH* v1.2 (Wlodzimierz *et al.* 2023), run with default parameters on the barn owl assembly v.4.0 (Machado, Cumer, *et al.* 2022) (more details in Supplementary Text 2 in Supplementary File 1). After identifying LGs, synteny with the chicken *reference genome* was inferred with the method presented in Waters *et al.* (2021). Synteny matches can be found in Supplementary Fig. 4 and Table 3 in Supplementary File 1. The rate of intrachromosomal shuffling was calculated from the genetic map distances following Supplementary information S5 in Veller *et al.* (2019). For every chromosome, there is a set of sequenced markers $A = (a_1, a_2,..., a_n)$ and a cumulative genetic distance up to each marker (in cM) of $D_a = \Sigma_{i < a} d_i$, but these markers are not placed at even distance intervals. Thus, we created a set of equidistant (10 kb) pseudomarkers $B$, starting from the beginning and including the end of the sequence, and calculated their centimorgan position by using adjacent true markers. For every $b$ marker, the distance ($D_b$) is:

$$D_b = \frac{s_{a(b)+1} - s_b}{s_{a(b)+1} - s_{a(b)}} D_{a(b)} + \frac{s_b - s_{a(b)}}{s_{a(b)+1} - s_{a(b)}} D_{a(b)+1}$$

where $s_{a(b)}$ is the genomic position of the nearest marker below marker $b$, $s_b$ the genomic position of marker $b$, and $s_{a(b)+1}$ is the genomic position of the nearest marker above $b$. Similarly $D_{a(b)}$ and $D_{a(b)+1}$ are the cumulative genetic positions of the nearest true markers below and above pseudomarker $b$, respectively. Then, we averaged all pairwise distances of the pseudomarkers, weighing the result by the proportion of length belonging to each LG (L2). The end value corresponds to the total shuffling attributable to each chromosome ($r_{intra}$).

Recombination rate estimates from pyrho were averaged across nonoverlapping windows of different lengths using a custom script. Windows of sizes 1 kb, 10 kb, 100 kb, and 1 Mb were created from the reference sequence using *bedtools makewindows* from *bedtools* v2.3 (Quinlan 2014). These windows were overlapped with the pyrho windows, and the recombination rate in centimorgans was calculated by multiplying the recombination probability estimate with the length of each interval and then translating this to centimorgans using Haldane's function (Haldane 1919). For each window, nucleotide diversity was calculated using *VCFtools* and the *--window-pi* command. Estimates were corrected for masked nucleotides in each window. Sequence GC content was calculated using the reference sequence and the *bedtools nuc* command. We annotated CGIs using the UCSC genome browser CGI annotation tool *cpg_hl* (Kent *et al.* 2002) with default parameters. The Gini coefficient was calculated using *Desctools* v.0.99 (Signorell 2023). Transcription start and end sites were annotated using the genome annotation from *NCBI* as the first and last positions of the genomic sequence for each gene. The intersection of different bed files was performed using bedtools. Local hotspots were annotated by dividing the estimate of recombination rate in each focal window with the average recombination in 80 kb around (40 kb upstream and 40 kb downstream). In addition, because power to infer hotspots is lower in very low and high recombination rates (Singhal *et al.* 2015), we restricted hotspots to windows between 1 and 10 cM/Mb. Lastly, we removed any hotspots found in windows with more than 500 bp annotated as repeats. Global hotspots were annotated as windows with at least 10 times the genome-average recombination rate.

Owl images are from *PhyloPic* (https://www.phylopic.org). Map was made using *tmap* v3.3-4 (Tennekes 2018) and the Natural Earth high-resolution data set. *Corrplot* v.0.92 was used for correlation (Wei and Simko 2021). *Vioplot* v0.4.0 was used to create the violin plots (Adler *et al.* 2022). All analyses were executed in R v.4.3.1 (R Core Team 2023) using the *Rstudio* IDE (Posit team 2022). Scripts with commands used for data generation and downstream analyses can be found in https://github.com/topalw/Recombination_barn_owl.

## Results

In order to build the most comprehensive set of polymorphism to date in the barn owl, we performed variant identification on a total of 502 whole genome sequences of medium to high coverage (mean 16X, range = 8X–43X). Samples originated from 19 distinct localities spanning the Western Palearctic distribution of the species (with 3–13 samples from 19 localities, see Supplementary File 2 for details). The Swiss population in particular included 346 individuals with a family structure originating from an observational pedigree. After filtering, we retained 26,933,469 single nucleotide variants (SNPs) and used subsets of those individuals and variants for each analysis below (Supplementary Table 1 in Supplementary File 1).

### LGs and recombination rate of the barn owl

After additional filtering on the variant set for technical errors, allele frequency, distance, and missingness (see *Methods*), we ordered 154,706 SNPs along the 41 largest scaffolds of the barn owl genome assembly v.4.0 (Machado, Cumer, *et al.* 2022) to create a linkage map for the species. Based on segregation of these markers in 350 meioses of 250 individuals from 28 families, we identified 40 LGs spanning 1,196.47 million base pairs (Mb) covering 95.7% of the genome assembly. All the LGs we identified correspond to scaffolds

in the genome assembly, except for Super-Scaffold 2, which was split into 2 LGs (see Supplementary Text 1 and Table 3 in Supplementary File 1). In addition, we merged Super-Scaffold_3 and Super-Scaffold_49 into LG 20 and Super-Scaffold_13 and Super-Scaffold_42 into LG 40, corresponding to the sex chromosome Z. The genome assembly of the barn owl therefore contains the sequence of 39 LGs out of 45 expected pairs of autosomal chromosomes and 1 LG corresponding to the Z chromosome. Of the 39 autosomal LGs identified, 17 correspond to microchromosomes—chromosomes with a size less than 24 Mb (Waters *et al.* 2021). After we were confident with the correspondence between scaffolds and LGs, we inferred the genetic map positions of a set of 4,889,667 SNPs, which were filtered with less stringent parameters, with an average inter-SNP nucleotide distance of 266 bp. This way we inferred the genetic map for each LG independently by capturing COs over most of the assembled sequence. We observed 3,972 male COs and 4,250 female COs, which correspond to 23 COs per meiosis on average. For the 39 autosomal LGs, the final sex-averaged linkage map length spanned 2,580 cM resulting in an autosomal genome-wide average estimate of recombination rate of ~2.33 cM/Mb for this species.

The genetic length of LGs increased slightly with their physical length (Fig. 1a). As a result of the obligate CO, the genetic map of a chromosome is expected to be at least 50 cM long. The LGs of the barn owl showed an average genetic length of 66 cM (range: 35–97 cM), and all LGs recombined on average less than twice per meiosis (<100 cM). The slope of the regression of the genetic length on the physical length was significantly positive [$\beta = 0.378$, 95% confidence interval (CI): 0.18–0.57], and the intercept was not different from the expected minimum of 50 cM ($\alpha = 55$, 95% CI: 48–62) under 1 obligate CO per chromosome. Three LGs around 30 Mb of physical length had an inferred genetic length less than 50 cM (LG13: 40 cM, LG18: 49.2 cM, and LG31: 34.8 cM, Supplementary Table 3 in Supplementary File 1).

We found that the Z chromosome is the largest chromosome, with an assembled physical length of 90.3 Mb (Fig. 1a; Supplementary Fig. 5, final panel, and Table 3 in Supplementary File 1). The identified pseudoautosomal region (PAR) spans 4.4 Mb at the end of the LG (Supplementary Fig. 5 in Supplementary File 1). Males, with 2 copies of the Z chromosome, had a genetic length of 69.28 cM (approximately 10 cM in the PAR), which corresponds to a rate of approximately 0.77 cM/Mb (Supplementary Table 3 in Supplementary File 1). Females, having a single copy of the Z chromosome, harbored 53 cM just in the PAR (Supplementary Table 3 and Fig. 5 in Supplementary File 1).

### Subtle heterochiasmy

To infer heterochiasmy, we compared the sex-specific linkage map estimates (Supplementary Fig. 5 and Table 3 in Supplementary File 1). Females, with a map length of 2,672 cM, had a 4.7% larger genetic map than males (2,560 cM). There appeared to be no consistent pattern of heterochiasmy among LGs at the chromosome scale (Fig. 1b). To investigate potential differences in localization of recombination events in males and females, we looked at the positioning of COs along the length of all LGs (Fig. 1c; Supplementary Figs. 5 and 6 in Supplementary File 1). Overall, COs occurred closer to the LG extremities than in their center. However, the distribution of COs differed between the sexes (two-sample Kolmogorov–Smirnov test $D = 0.30$; $P < 0.001$). Notably, males appeared to recombine more at the extremities and the middle of the LGs since their COs were less evenly distributed.

To test if pericentromeric rates differ between males and females, we attempted to annotate centromeres in silico in the

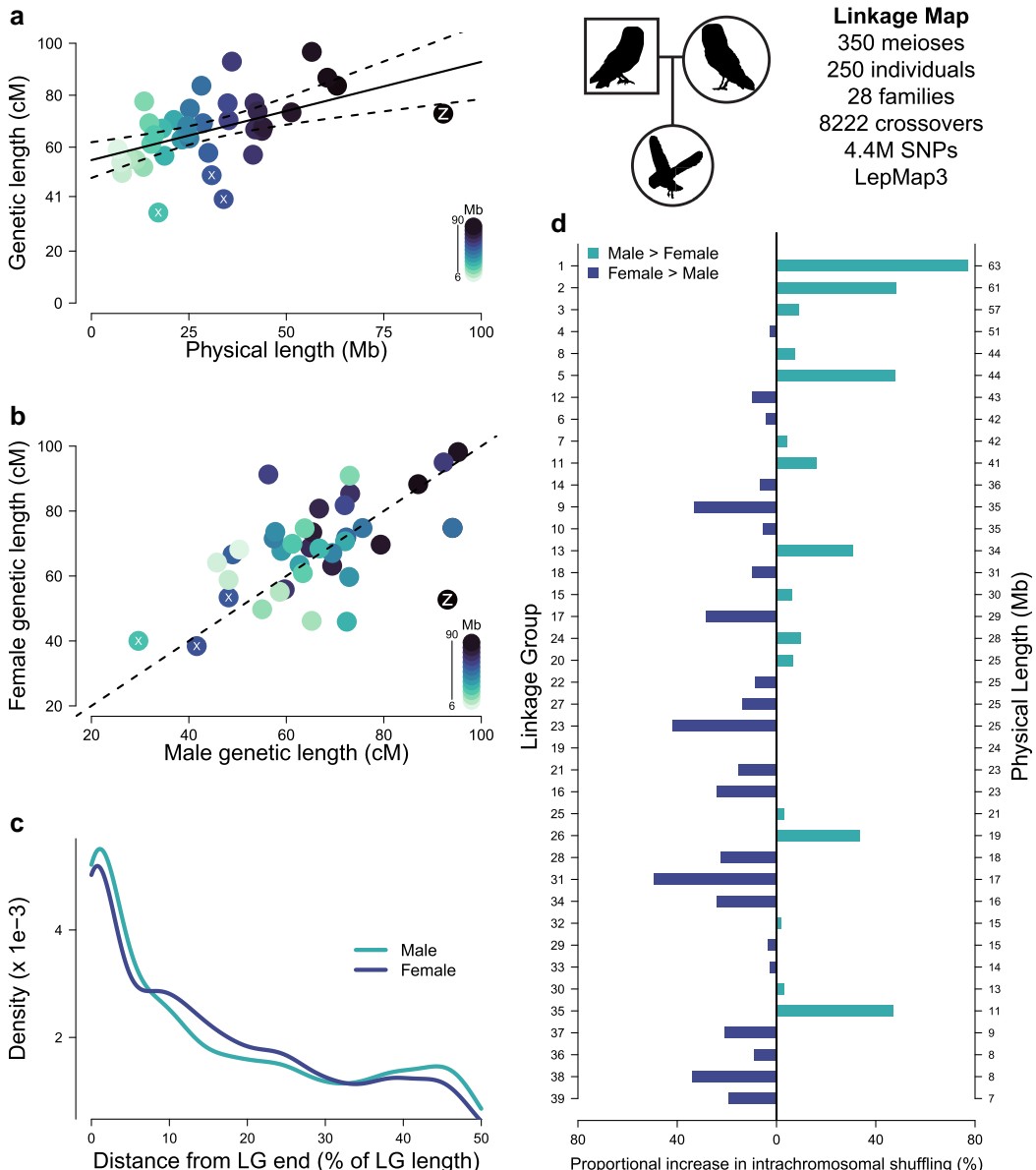

**Fig. 1.** Linkage map for the barn owl (*T. alba*). All plots in this figure are products of the linkage map data set consisting of 350 meioses of 250 individuals in 28 families, illustrated with the pedigree of owl symbols on the top right. a) Estimates of sex-averaged genetic lengths for the LGs plotted against their physical length. Regression line is shown with 95% CIs ($\alpha = 55$, $\beta = 0.378$, $t = 3.73$, $P < 0.001$). Color intensity scales with LG physical lengths. The Z LG is marked with a Z, and LGs with less than 50 cM are marked with a white X. b) Female map length (cM) of LGs against the male map length. Dashed line is the identity ($y = x$) line. c) Density plot of male (aquamarine) and female (blue) CO counts plotted along the distance from the LGs' end. X-axis is in percentages of total LG sequence. Density values are scaled so that they sum to 1. d) Differences between sexes in rates of intrachromosomal shuffling ($r_{intra}$) presented as a ratio (male $r_{intra}$/female $r_{intra}$) for different LGs. Bars to the right of the black line colored green signify higher intrachromosomal shuffling in males, and bars to the left of the black line colored in blue correspond to LGs with higher shuffling in females. LGs are ordered by decreasing physical length as on the right-hand axis.

barn owl assembly. Annotated tandem repeats were often concentrated at the ends of chromosomes as expected by the presence of a centromere in acrocentric chromosomes. We retained only annotated centromeres in 18 LGs, where the centromere was inferred in the distal 20% of the sequence. However, most annotated centromeres overlapped recombination peaks, and we were not fully confident in the end result to draw any conclusions (Supplementary Text 2 and Figs. 7 and 8 in Supplementary File 1).

Because sexes showed different locations of COs and not all COs are as effective at shuffling alleles between haplotypes, we quantified the rate of intrachromosomal shuffling ($r_{intra}$) as defined in Veller *et al.* (2019). Briefly, this quantity measures the

relative shuffling of alleles due to a CO along the length of the chromosome. For example, a CO in the middle of the chromosome shuffles more alleles than a distal one. We estimated rates of intrachromosomal shuffling in males and females (Fig. 1d). Despite an overall lower recombination frequency (Fig. 1b), males showed up to 50% higher intrachromosomal shuffling for larger LGs (Fig. 1d). On the other hand, females showed higher rates of shuffling in intermediate to smaller LGs.

## Fine-scale variation among LGs

To investigate fine-scale variation in recombination rates, we turned to recombination rates estimated from patterns of LD

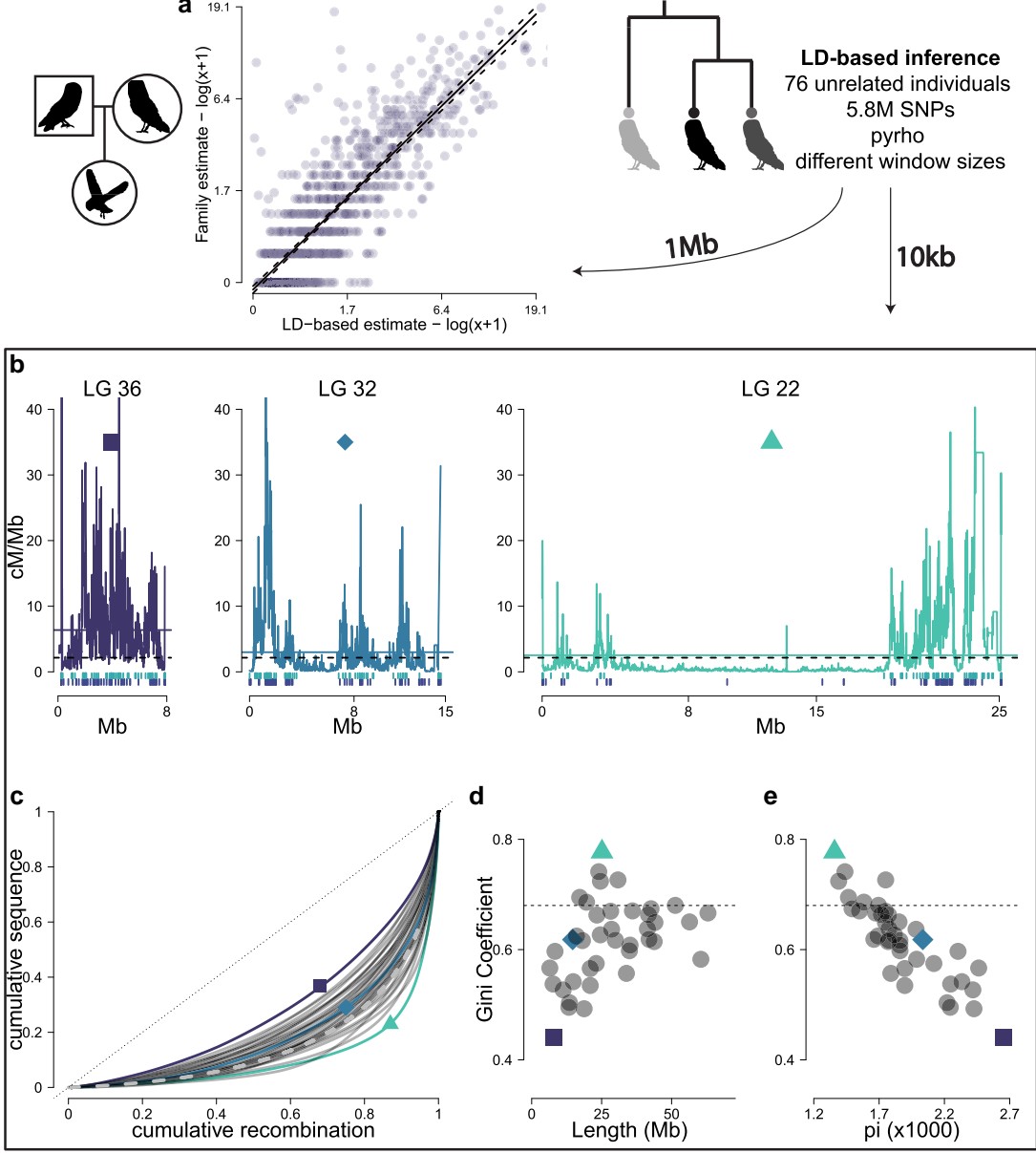

**Fig. 2.** Variation of recombination among LGs. a) Comparison of recombination rate estimates from linkage mapping and LD inference for the Swiss population. The comparison is made in 1 Mb windows. Regression line is shown with prediction intervals as dashed lines ($\alpha = -0.08$, $\beta = 0.999$, $t = 53.15$, $P < 0.001$). b) The recombination frequency (cM/Mb) in 10 kb windows along the physical map of 3 example LGs: LG 36 (purple square), LG 32 (aquamarine diamond), and LG 22 (green triangle), respectively. The dashed horizontal line represents the genome-average recombination rate and the full line the LG average recombination rate. These LGs represent LGs with different recombination landscapes. Vertical lines below indicate locations of male (top row) and female (bottom row) COs from the family estimates. c) Cumulative sequence plotted against cumulative ordered recombination length for each LG (dark gray curves) and genome-wide (gray dashed curve). The black dotted line is the identity ($y = x$) line. The Gini coefficient corresponds to the area delimited by each curve and the identity line. d) The Gini coefficient of recombination rates for each LG plotted against its physical length. Dashed gray line is the genome-average Gini coefficient. e) The Gini coefficient of recombination plotted against the average nucleotide diversity of each LG.

(Supplementary Fig. 9 in Supplementary File 1). We estimated recombination rates using *pyrho* v0.1.7 (Spence and Song 2019) in a set of 5.8 million variants (number of variants per population is presented in Supplementary Table 1 in Supplementary File 1) identified along the whole genome of 76 unrelated birds from Switzerland (CH). The total genetic length for the autosomal part of the assembly was estimated from LD to be ~3,100 cM, 25% larger than the linkage map estimate for the same population. We scaled the total length inferred from LD to be the same as the linkage mapping estimate for each scaffold separately, to account for the confounding effect of $N_e$ and compared the estimates in non-overlapping 1 Mb windows. The correlation of recombination rate estimated from the linkage map and from *pyrho* at the 1 Mb scale was high ($r = 0.83$, 95% CI: 0.813–0.85), but *pyrho* showed larger estimates in regions of low recombination compared to linkage mapping (Fig. 2a).

The recombination landscape differed among chromosomes. To further quantify this variation in recombination rates, we calculated the proportion of genomic sequence where recombination occurs. We ordered all 10 kb windows for each LG by decreasing recombination rate and quantified the cumulative recombination percentage against the cumulative percentage of sequence

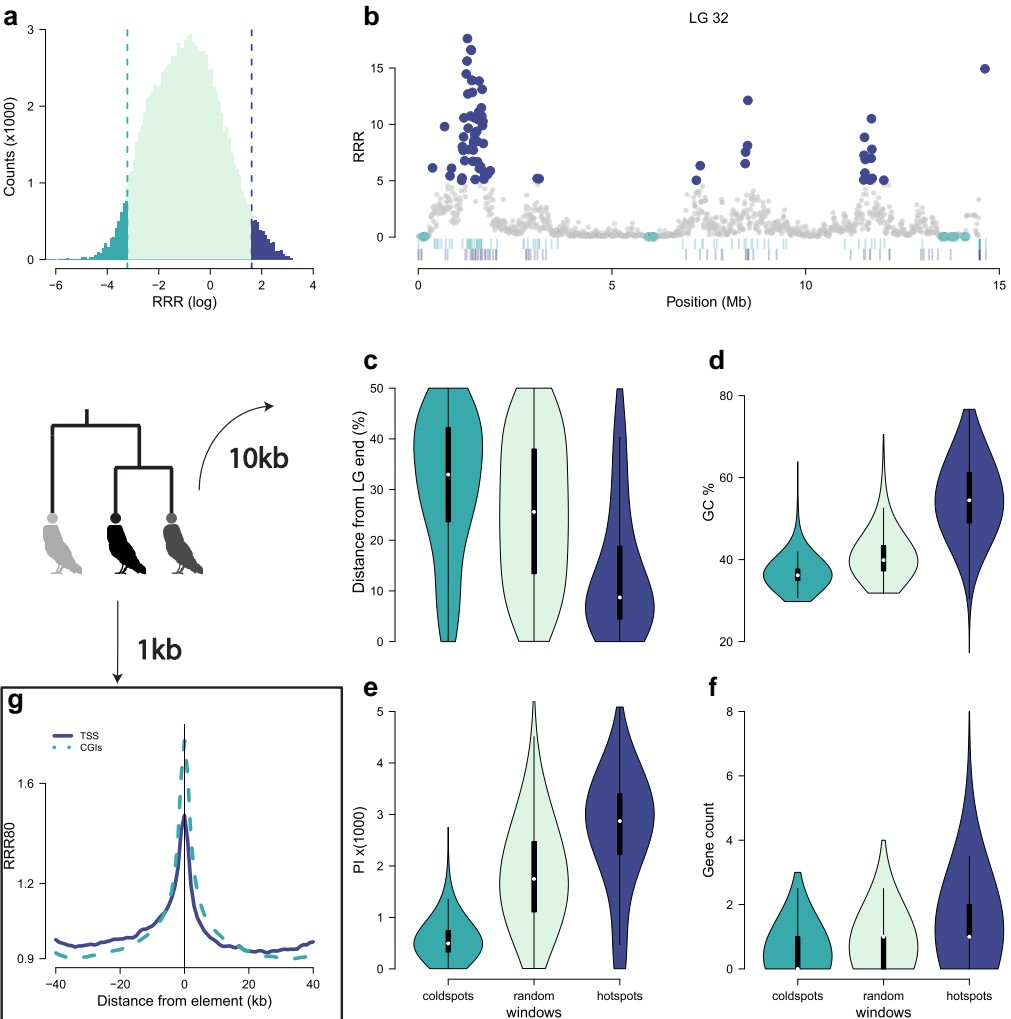

**Fig. 3.** Correlates of recombination. a) The distribution of recombination rates in 10 kb windows. We define hotspots as windows with more than 5 times the genome-average recombination rate and coldspots as windows with a recombination rate less than a fifth of the average. b) Example of annotated coldspots and hotspots in LG 32. Vertical lines below are locations of male (top row) and female (bottom row) COs from the family estimates. Violin plots of characteristics of hotspots, coldspots, and a similar sized set of random windows that belong in neither category. c) Position along LG. d) GC content. e) Nucleotide diversity. f) Number of annotated genes. g) Recombination is increased around annotated TSSs (full line) as well as CGIs (dashed line). RRR80 is the recombination rate of each 1 kb window divided by the average in 80 kb around. The lines show the average across all identified elements.

(Fig. 2c). Overall, 80% of recombination occurs in approximately 30% of the sequence (dashed gray line in Fig. 2c). However, there was substantial variation in the distribution of recombination among LGs. To further measure this skewness, we used the Gini coefficient of recombination rates for each chromosome, which is a measure of inequality among the values of a frequency distribution. Graphically, this corresponds to the area between each curve in Fig. 2c and the identity ($y = x$) line and ranges from 0 to 1. Smaller values indicate an evenly spread landscape (every window has the same recombination rate) and higher values a more variable one (windows show large differences in recombination rates). The genome-wide average Gini coefficient was 0.67, and the LG-specific estimates varied between 0.44 and 0.77 (LG 36 and LG 22, marked with a blue square and a green triangle, respectively, in Fig. 2). Along with a LG of an intermediate Gini coefficient of 0.61, LG 32 (aquamarine diamond in Fig. 2), their recombination landscapes are presented in Fig. 2b. We found that the Gini coefficient depended on the physical length of the LG, with more evenly spread (and elevated) recombination rates in smaller LGs and more concentrated landscapes in larger

ones (Fig. 2d), although the effect reached a plateau as the length increased above 25 Mb. Further, the Gini coefficients were strongly negatively correlated with the average nucleotide diversity of the LG, with more concentrated recombination peaks associated with lower average nucleotide diversity (Pearson's $r = -0.87$, 95% CI: $-0.92, -0.76$) (Fig. 2e). Overall, recombination rates varied substantially among the different LGs of the barn owl assembly.

## Correlates of recombination

To identify genomic correlates of recombination rates, we annotated 10 kb windows with more than 5 times the genome-wide average recombination rate as recombination "hotspots" windows (Fig. 3a). We identified 3,805 such windows containing 30% of the total genetic length. We also annotated 4,677 windows with less than 5 times the average recombination rate, i.e. recombination "coldspots." We compared both hotspot and coldspot with a set of 4,000 randomly chosen windows that were not annotated as either hotspots or coldspots. An example of hotspots and coldspots across a particular LG is shown in Fig. 3b. We found that windows

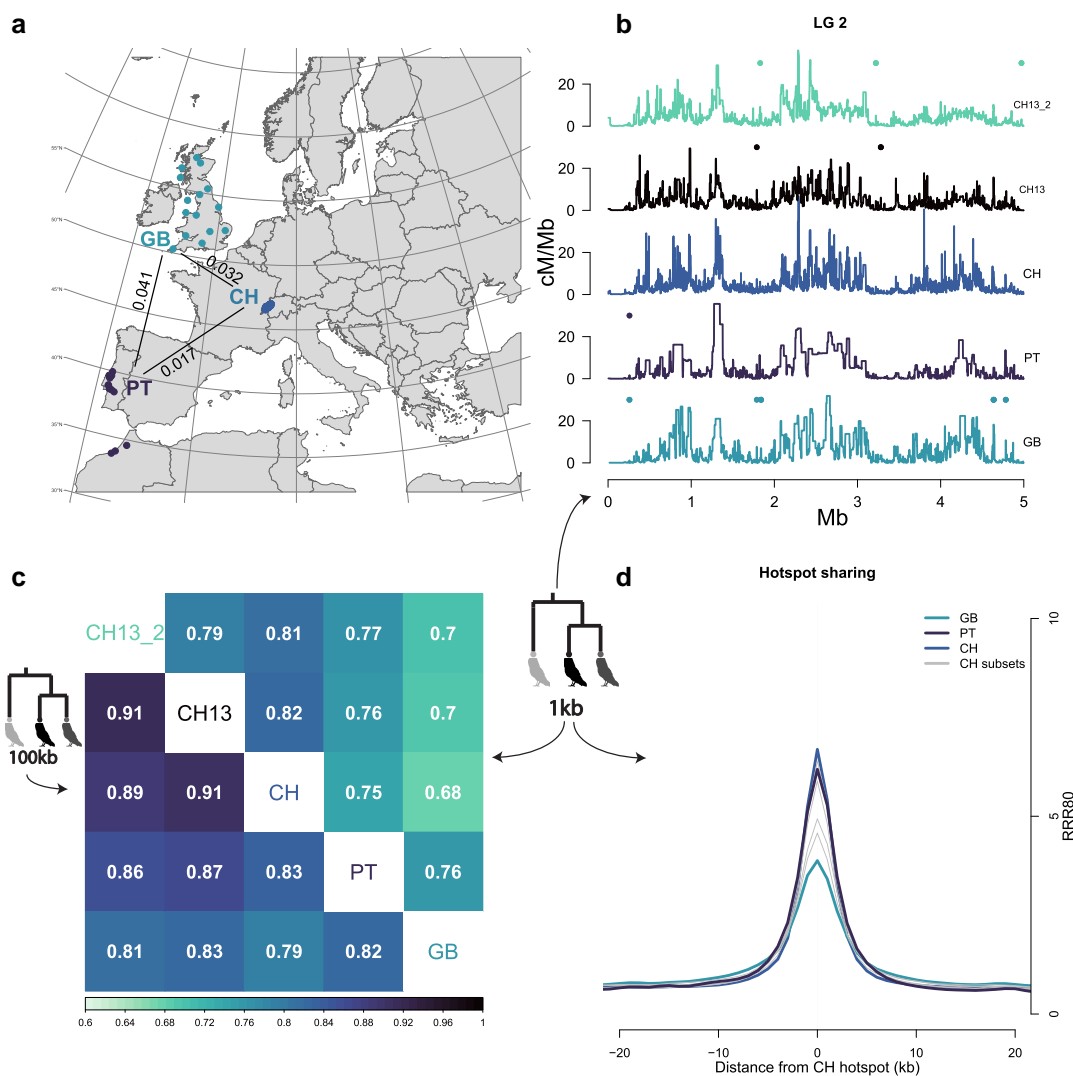

**Fig. 4.** Comparison of recombination landscapes between populations. a) Map of sampled populations. Numbers correspond to genome-wide pairwise FST values (Machado, Cumer, et al. 2022). b) Example of recombination landscape for 1 kb windows in all data sets for the first 5 Mb of LG 2. Points above each plot show local hotspots for each population. Local hotspots are identified as at least two consecutive windows with a recombination rate of at least five times the average in a region of 80kb around them (RRR80 > 5). c) Correlation matrix of recombination rates for 1 kb windows (above diagonal) or 100 kb windows (below diagonal). d) Relative recombination rates in 80 kb around (only 40 kb plotted) 1 kb windows annotated as hotspots in the full Swiss data set. Each line is the relative recombination rate in all other populations. Gray lines are the subsets from Switzerland with $n = 13$. CH: full Swiss data set ($n = 76$); PT: Portuguese data set ($n = 13$); GB: Great Britain data set ($n = 13$); CH13: undersampled first Swiss data set ($n = 13$); CH13_2: undersampled second Swiss data set ($n = 13$).

with high recombination rates were most often identified at the ends of LGs, had higher GC content and nucleotide diversity, and colocalized with genes (Fig. 3c–f). On the contrary, coldspots were found in the middle of the LGs and were depleted in GC content, nucleotide diversity, and genes.

In addition, because birds lack the *PRDM9* gene, recombination hotspots are expected to localize to TSSs, as well as CGIs (Singhal *et al.* 2015; Baker *et al.* 2017). To verify this, we used estimates of recombination frequency in non-overlapping windows of 1,000 bp (1 kb) along the genome. Windows that were annotated to contain a TSS ($n = 17{,}191$, 1.5% of windows) or contained a CGI spanning the whole window ($n = 13{,}971$, 1.3% of windows) were identified, and their recombination rate was divided by the average recombination rate in 40 kb upstream and 40 kb downstream of the focal window (relative recombination rate in 80 kb—RRR80). The focal windows showed elevated recombination rates compared to other windows in their vicinity (Fig. 3g).

## Recombination landscapes across populations

To quantify the change of the recombination landscape across European populations of the species, we used 3 populations from the Western Palearctic: Portugal (PT, $n = 13$), Great Britain (GB, $n = 13$), and Switzerland (CH, $n = 76$) (Fig. 4a). For Portugal, we pooled together 3 samples from Morocco and 10 from Portugal since they have very high genetic similarity (Cumer, Machado, Dumont, *et al.* 2022). Because the sample size in Switzerland was far larger than the other 2 populations and to test the robustness of our results, we randomly subsampled 5 sets of 13 individuals from Switzerland, creating pseudoreplicate populations. Genetic length estimates differed between populations. Portugal showed a 5-fold increase in genetic length while GB showed a 2-fold increase compared to the family linkage map. The Swiss subset data sets underestimated the total genetic length up to 2.4 times. For all populations, we scaled the results so that each scaffold would match the corresponding linkage map

estimate. After scaling, we compared all populations with the linkage map in 1 Mb windows along the genome and found that genome-average correlations were higher than 0.69 (Supplementary Fig. 10 in Supplementary File 1). At a finer scale, however, recombination landscapes varied among populations (an example landscape for all populations is presented at the 1 kb scale for the first 5 Mb of LG 32 in Fig. 4b; all landscapes in Supplementary Fig. 11 in Supplementary File 1). Estimates in GB showed reduced resolution at the finer scales (aquamarine line in Fig. 4b), likely due to reduced genomic diversity and historically smaller $N_e$ in the population (Supplementary Table 1 and Fig. 3 in Supplementary File 1; see also Machado, Cumer, *et al.* 2022).

We quantified the divergence of the recombination landscapes as the correlation of recombination rates on different window sizes (1 and 100 kb) among all pairs of populations (Fig. 4c). The recombination landscapes between populations were concordant at the broad scale with the correlation steadily decreasing as the window size got smaller. The subsampled sets from Switzerland showed that correlations between CH and PT were slightly lower than between CH and the subsets. It also showed that broad-scale correlations in GB were not substantially no different between the CH subsets and PT while the fine scale (1 kb) was not as well correlated.

Figure 2b and c provide evidence for the existence of fine-scale hotspots in the barn owl recombination landscape. Thus, we looked for at least 2 consecutive 1 kb windows with a recombination rate higher than 5 times the average in 80 kb around them (relative recombination rate in 80 kb—RRR80), following definitions of a local hotspot in the literature (Myers *et al.* 2005; Singhal *et al.* 2015; Kawakami *et al.* 2017), and careful filtering based on recombination rates and repeat content. After filtering, a set of 1,566 windows were identified as local hotspots, harboring 8,171 variants. We then quantified the RRR80 of the other populations on those windows (Fig. 4d). The 5 Swiss subsampled data sets with $n = 13$ provided a measure of estimation error and showed strong to moderate increase on the hotspot windows. Recombination rates in the Portuguese sample showed a marked increase in the windows, indistinguishable from the Swiss subsampled data sets, while the signal in the samples from Great Britain was outside the range of the subsampled data sets.

## Discussion

Recombination is an evolutionary force with direct and indirect implications for reproduction and evolution, and while it varies on different scales, our knowledge and ability to quantify this variation are often limited. In this study, using an extensive whole genome sequencing data set of barn owls, we inferred recombination with 2 different methods to describe broad and fine-scale variation in recombination variation: linkage mapping on a pedigreed population and a LD-based approach on 3 different populations. Using both methods allowed us to identify crossing over in contemporary families and infer haplotype shuffling in the coalescent history of our data set. We first identified 40 LGs in the barn owl genome assembly and estimated the recombination rate to be ~2.3 cM/Mb. We showed that in the barn owl, the overall length of genetic maps shows no clear distinction between males and females, but sexes had fine-scale differences in CO placement and shuffling proportions. Despite few (1 to 2) COs per chromosome, we found large variation in the location of these COs among different LGs, with some LGs harboring large stretches of reduced recombination. We showed that this variation in the distribution of recombination comes in tandem with variation in the genetic

diversity of the different LGs. At a more fine-scale resolution, recombination rates were increased in windows that contain TSSs and CGIs. Equally, recombination hotspots at the 10 kb scale showed an elevated GC ratio, diversity, and gene density and were most often found at the ends of LGs. Lastly, population comparisons showed local recombination hotspot conservation despite high statistical noise. We discuss these results and their implications below.

## LGs confirm the near completeness of the barn owl assembly

Complete genome assemblies are a useful resource that requires multiple sources of information. The karyotype of the barn owl contains 45 autosomal pairs with a large Z chromosome and degraded W (Belterman and De Boer 1984; Rebholz *et al.* 1993; Peona *et al.* 2018). The latest barn owl assembly v4.0 (Machado, Cumer, *et al.* 2022) was assembled into superscaffolds using optical genome mapping (BioNano; Lam *et al.* 2012). In the present study, we verified and improved the barn owl assembly by anchoring the largest 41 scaffolds into 40 LGs and revealed that the genome assembly of the barn owl is of chromosome-level quality (though not telomere to telomere). However, there are still 6 autosomes unaccounted for in the linkage map, and the W chromosome is also not present because the assembly uses a male individual (Ducrest *et al.* 2020; Machado, Cumer, *et al.* 2022). These small autosomes might be partially present in the physical assembly, since smaller scaffolds with a few tens of identified markers that passed filtering could not be confidently allocated to LGs. Regardless, the "missing" chromosomes are likely the smallest 6 microchromosomes, or dot chromosomes, notoriously difficult to sequence and assemble due to their high GC content and reduced chromatin accessibility (Burt 2002; Bravo *et al.* 2021; Waters *et al.* 2021). Notably, the dot chromosomes were only recently assembled in the chicken genome (Huang *et al.* 2023) and are missing from most available bird reference genomes (Peona *et al.* 2018; Baalsrud *et al.* 2024). It is likely that most microchromosomes will remain elusive until future studies make use of advances in long-read technologies (Marx 2023) to complete the reference genomes of birds. In this endeavor, linkage mapping, when available, can be a valuable tool (e.g. Peñalba *et al.* 2020; Robledo-Ruiz *et al.* 2022).

## Differential shuffling between sexes due to CO placement

Our results suggest the presence of variation in CO placement between sexes that is not immediately apparent when investigating sex differences at overall genetic map length of each LG. This variation generates a differential shuffling of markers in each sex among different chromosomes with the overall pattern showing that males recombine less evenly than females along the length of the LGs. In turn, this leads to a greater shuffling of markers, as observed in larger chromosomes, where male COs situated more toward the middle of the sequence shuffle most of the variants in the chromosome. In avian studies, results are inconclusive for a general pattern of heterochiasmy in the class. For example, male collared flycatchers and male hihis (*Notiomystis cincta*) exhibit higher genetic lengths than females and recombine more toward the ends of the chromosomes (Kawakami *et al.* 2014; Smeds *et al.* 2016; Tan *et al.* 2024). On the other hand, sparrows (*Passer domesticus*) and great tits (*Parus major*) show higher female recombination (van Oers *et al.* 2014; McAuley *et al.* 2024), and other species such as the great reed warbler (*Acrocephalus arundinaceus*) show no pattern on the broad scale but a clear male bias toward

the telemeres (Zhang et al. 2024). Recently, there has been an effort to recharacterize heterochiasmy in CO placement with studies showing that there are heritable fine-scale sex differences in recombination in birds (McAuley et al. 2024; Tan et al. 2024; Zhang et al. 2024). Although fine-scale information on sex differences is not available for other bird species, the emerging pattern is that descriptions of recombination patterns at the LG level might not reveal the whole picture of heterochiasmy in birds and a more thorough quantification of sex-specific recombination is required.

Heterochiasmy is pervasive in eukaryotes (Burt et al. 1991; Sardell and Kirkpatrick 2020). In the most extreme version of sex differences, one sex does not recombine at all (achiasmy), almost always the heterogametic sex [for example XY *Drosophila* males and ZW females in Lepidoptera or the common silk worm (*Bombyx mori*); Morgan 1914; Burt et al. 1991; Goldsmith et al. 2005; Jiggins et al. 2005]. For most species, the reality is more nuanced and is independent of which sex is heterogametic. A prominent pattern of heterochiasmy is that males often recombine more toward the telomeres of a chromosome (Kong et al. 2010; Giraut et al. 2011; Johnston et al. 2016; Brekke et al. 2023; Venu et al. 2024), although there are exceptions (Kianian et al. 2018; Rifkin et al. 2022). While the evolutionary reasons behind the existence of heterochiasmy remain unexplained (Burt et al. 1991; Mank 2009; Sardell and Kirkpatrick 2020), hypotheses have been proposed such as differences in haploid selection intensity (Lenormand and Dutheil 2005) or female meiotic drive (Brandvain and Coop 2012). Haploid selection predicts that the sex with more intense selection during the haploid phase (e.g. sperm competition) will show reduced recombination while the meiotic drive hypothesis predicts increased rates of recombination close to the centromeres in females. Barn owls show very low rates of extrapair copulation (Roulin et al. 2004) and therefore low sperm competition, and in birds and mammals, the egg completes meiosis II only after fertilization leaving little room for haploid selection (Mira 1998). Further, the emerging patterns of heterochiasmy, and its frequent absence, do not lend support to the haploid selection theory in birds, which was also not associated with sperm competition intensity in eutherian mammals (Mank 2009). Concerning female meiotic drive, because most avian assemblies lack annotation of centromeres, thus far no studies have been published on this topic in birds. While we attempted to annotate centromeres in our assembly, the lack of assembled repeats (often the last part of a reference to be assembled) and the placement of centromeres at the distal ends of the LGs confounded results (Belterman and De Boer 1984; Rebholz et al. 1993; Benham et al. 2024). Thus, to reach conclusions about heterochiasmy in birds, the use of more complete reference genomes with assembled repeats will be invaluable (Peona et al. 2018; Robledo-Ruiz et al. 2022; Huang et al. 2023).

Heterochiasmy is most apparent on the sex-linked chromosome. Females, the heterogametic sex in birds, recombine exclusively on the PAR of the Z chromosome where sequence homology between the Z and W chromosomes is maintained. Beyond the PAR region, the 2 chromosomes have diverged so drastically in the >100 million years since their evolution from a pair of autosomes, that in karyotypic studies they appear like different elements (i.e. they are heteromorphic) (Belterman and De Boer 1984; Rebholz et al. 1993; Handley et al. 2004). While this is true for most birds (Neognathae), ratite birds (Palaeognathae) maintain homomorphic sex chromosomes with large PAR, which spans 50 Mb in the ostrich (Mank and Ellegren 2007; Yazdi et al. 2023). The variable degradation of sex chromosome pairs has been linked to multiple factors, like age of the sex chromosomes and strength of sexual conflict, in animals and especially in plants that have evolved ZW and XY systems independently multiple times (Bergero and Charlesworth 2009; Charlesworth 2013; Wright et al. 2016; Charlesworth 2019). Future work in the barn owl such as assembling the W chromosome and identifying the sex-determining region and its contents will help shed more light on the place of the Tytonidae on the sex chromosome evolution landscape.

## Variation in recombination rate among and within LGs

Our results show that barn owl LGs recombine at most twice per meiosis. This result is in line with an expectation of 1 CO per chromosome (or chromosome arm) and the generally small (<70 Mb) acrocentric (or telocentric) chromosomes in the barn owl karyotype (Belterman and De Boer 1984; Rebholz et al. 1993; Coop and Przeworski 2007). However, these recombination frequencies contrast with results from other bird species. For example, the syntenic chromosome 2 of chickens and flycatchers with an approximate length of 150 Mb shows an average of 6 COs per meiosis (300 cM) (Groenen et al. 2009; Kawakami et al. 2014). The same chromosome has a genetic length of 175 cM in sparrows and 100 cM in great tits and superb fairy-wrens (van Oers et al. 2014; Peñalba et al. 2020; McAuley et al. 2024). The source of this variation in the order is unknown. Reasonable hypotheses include the evolution of the recombination landscape, localized suppression of recombination in some species (for example through segregating structural variations like inversions), or interspecific variation in the strength of CO interference (Kirkpatrick 2010; Otto and Payseur 2019). Higher quality linkage map data on more bird species can help identify the breadth of recombination variation in the class, and a meta-analysis of the available data sets can provide a much needed rigorous comparison.

In the barn owl, some LGs appear to recombine less than once per meiosis (genetic length < 50 cM). However, since the absence of an obligate CO can lead to aneuploidy, which, coupled with the LGs' intermediate size, should generate severely deleterious consequences, this is unlikely to be the true recombination frequency of these LGs (Hassold et al. 2007). A more likely explanation is that in these chromosomes, the genetic length is less than 50 cM due to missing markers at the distal parts of the chromosome. Because our marker data set used is relatively extensive, this discrepancy likely originates at the assembly stage, where parts of the sequence might have been misassembled or be present in small scaffolds. Another notable outlier in our study is the Z chromosome that in males recombines almost 3 times less than the autosomes. One explanation is a lack of marker coverage, especially for scaffold 13, that might have been filtered out during linkage mapping. On the other hand, the Z chromosome harbors a low-recombination region in the middle and is the only metacentric chromosome in the barn owl karyotype and the reduction could be due to pericentromeric suppression or segregating structural variations that are found on the Z chromosome in other bird species (Knief et al. 2016, 2017; Yazdi and Ellegren 2018).

We find recombination rates to vary substantially between and within chromosomes. As expected from an obligate CO per chromosome, smaller chromosomes tend to have higher rates (per bp) of recombination compared to longer chromosomes. In addition, smaller chromosomes show a more uniform distribution of recombination rates along their length. Longer chromosomes exhibit a U-shaped pattern, with reduced recombination in their center sometimes spanning large parts of the chromosome.

This relationship between and within chromosomes has been found in other species (e.g. Backström *et al.* 2010; Bascón-Cardozo *et al.* 2024; Castellani *et al.* 2024), and Haenel *et al.* (2018), in a meta-analysis of recombination rates of different species, proposed a model where the length of a chromosome and the distance from the telomere are the major factors impacting recombination rates. This model was recently extended by Brazier and Glémin (2022) based on a large data set of plant linkage maps, to include centromeric position and the placement of a single CO per chromosome. Our results follow this general pattern, although most LGs have interspersed signals of reduced recombination that could be due to segregating structural variants and/or CO interference (Kirkpatrick 2010; Otto and Payseur 2019). Regardless, the placement of COs on distal parts of the chromosomes is an important aspect of recombination that can inform different genetic analyses, from calculation of summary statistics to selection inference (Knief *et al.* 2017; Booker *et al.* 2020).

In the barn owl, these broad-scale patterns of recombination variation are associated with varying levels of nucleotide diversity, a correlation which we observe on 2 scales. Small LGs with uniform and large recombination rates show higher genetic diversity than large LGs with punctuated landscapes. At the same time, at a finer scale (10 kb), hotspot windows coinciding with family COs show higher nucleotide diversity than coldspot windows where family COs are rare. A similar correlation of genetic diversity and recombination is observed in bird species and across the tree of life, although there are exceptions and its magnitude varies (Nordborg *et al.* 2005; Webster and Hurst 2012; Cutter and Payseur 2013; Kawakami *et al.* 2014; Rowan *et al.* 2019; Peñalba *et al.* 2020). A proposed explanation for this association is the mutagenic effect of recombination that has been shown to exist in multiple species (Arbeithuber *et al.* 2015; Rattray *et al.* 2015; Halldorsson *et al.* 2019). Another explanation is the interplay of selection and recombination. If recombination is spread throughout the length of the sequence, neutral alleles are uncoupled faster from selected ones that tend to drag them to extinction or fixation, thus allowing an increase of standing variation. On the contrary, long stretches of reduced recombination, through the action of linked selection, lead to reduced diversity (Charlesworth *et al.* 1993; Charlesworth and Jensen 2021). While both effects can be acting at the same time, it can be hard to distinguish their relative contributions especially because recombination-induced mutation is hard to quantify in nonmodel species and the relative effects might be species specific (Cutter and Payseur 2013).

A recurring observation is that recombination rates correlate with multiple genomic features without a clear direction of causality. For example, as illustrated here, windows with increased recombination rates are found in gene-rich regions, a finding prevalent in both plants and animals (e.g. Paape *et al.* 2012; Kawakami *et al.* 2014; Rifkin *et al.* 2022). An attractive explanation for this pattern is an adaptive one. In gene-rich regions, targets for selection (positive or negative) are increased, and as a consequence, so is the potential for interference between selected alleles [Hill–Robertson interference (HRI); Hill and Robertson 1966]. Higher recombination rates in these regions can mitigate this effect. However, indirect selection for increased recombination might be too weak to drive this phenomenon (Roze 2021), since only a few COs are required to overcome HRI. Instead genes and COs might colocalize because of underlying factors like accessible chromatin, as suggested here by the increase of recombination around TSSs and CGIs or through modulating effects of transposable element density (Baker *et al.* 2017; Kent *et al.* 2017;

Kianian *et al.* 2018; Venu *et al.* 2024). Similarly, GC content can correlate with increased recombination both through colocalization of COs and CGIs and through gBGC (Eyre-Walker 1993; Duret and Galtier 2009). As recombination landscapes of more species accumulate, comparative studies will be more equipped to answer questions regarding the directionality and strength of these forces. For example, characterizing recombination patterns in species with varying magnitudes of these forces (e.g. different GC conversion bias or different strength of recombination-induced mutagenesis) might help pick apart their relative contribution. To this end, exploring the recombination landscapes of different species is a valuable first step.

## Population differences

The populations we use in our study show a shallow genetic differentiation resulting from an out of refugium expansion following the last glacial maximum and facilitated by moderate dispersal over the Western European continent (Cumer, Machado, Dumont, *et al.* 2022; Machado, Topaloudis, *et al.* 2022). Inspired by Talbi *et al.* (2024), we attempted to measure statistical noise from the sampling of genealogies in the inference of recombination landscapes through subsampling the Swiss data set 5 times down to $n = 13$. This showed that while correlations between Switzerland and Portugal were slightly lower than between Switzerland and the subsets, hotspot sharing was indistinguishable. We interpret this as evidence of divergence in parts of the recombination landscape. While this can be a signal of between-population differences in segregating structural variants, transposable elements, or variation in some part of the recombination machinery, it can also be confounded through the effect of LD. Inferring recombination rates through LD means that selection and population size fluctuations can impact the inferred result (O'Reilly *et al.* 2008; Johnston and Cutler 2012; Dapper and Payseur 2018). Especially at the scale of a few kilobases, methods are known to show large statistical noise that depends on sample size, demography, and sequencing artifacts (Raynaud *et al.* 2023; Talbi *et al.* 2024). In our study, estimates were especially problematic with the samples from Great Britain where low genetic diversity coupled with a small sample size led to inconclusive results. While LD-based approaches provide an easy way to quantify the recombination landscape in multiple populations or species, ruling out confounding factors can be challenging, unless a more robust method of inferring recombination is used, such as linkage mapping.

Concerning evolving recombination landscapes, past investigations have focused on the effect of the presence and absence of the PRDM9 gene. Species where the gene is active seem to have fast-evolving hotspots, possibly through erosion of identified motif sites (Coop and Myers 2007; Myers *et al.* 2010; Baker *et al.* 2017; Raynaud *et al.* 2024). On the contrary, in species without PRMD9, recombination hotspots tend to be preserved over longer evolutionary scales and colocalize with open chromatin (Auton *et al.* 2013; Lam and Keeney 2015; Singhal *et al.* 2015; Kawakami *et al.* 2017). Recently, both the relevance of PRDM9-driven hotspots (Hoge *et al.* 2024; Joseph *et al.* 2024) and the evolutionary stability of recombination landscapes in species without the gene have been questioned (Talbi *et al.* 2024). Beyond mammals, the lack of a fast-evolving hotspot gene does not imply the lack of recombination divergence that has been observed even in closely related taxa (Stapley *et al.* 2017). Multiple genes have been shown to harbor heritable variation in modifying the number and placement of CO events and have thus the potential to evolve under direct and indirect selective pressures (Reynolds *et al.* 2013;

Johnston *et al.* 2018; Dreissig *et al.* 2020; Arter and Keeney 2024 ; Johnston 2024). Yet, how often differences in recombination landscapes are adaptive is less well known.

## Conclusion

To conclude, we present the recombination landscape of the barn owl using both linkage mapping and LD-based inference. The barn owl genome is now equipped with an assembly comprised of 40 identified distinct LGs and a detailed recombination map. It is thus the first species in the Strigiformes order with significant genomic resources, paving the way for further analyses like genome-wide association studies and haplotype phasing. Concerning recombination, the 2 methods applied allow us to quantify variation in recombination between populations, sexes, chromosomes, and fine-scale genomic windows. We verify that observations in passerine species, like fine-scale heterochiasmy and large regions of no recombination, are found outside of this clade. We also highlight the complex interplay of recombination and genetic diversity. Overall, these results contribute to our growing understanding of recombination in eukaryotes and birds specifically, providing a more comprehensive overview of the changing recombination landscape and divergence between sexes.

## Data availability

All sample information is found in Supplementary File 2. Sequence data used in the study from previous publications are available on NCBI under BioProject codes PRJNA700797, PRJNA727915, PRJNA727977, PRJNA774943, and PRJNA925445. Sequence data generated for this study are available on NCBI under BioProject code PRJNA1172395. Code to reproduce the figures can be found on github: https://github.com/topalw/Recombination_barn_owl. Downstream data to be used with aforementioned scripts can be found on Zenodo under the following DOI: 10.5281/zenodo.13982582.

Supplemental material available at GENETICS online.

## Acknowledgments

The authors would like to thank Museum national d'Histoire naturelle in Paris for the Corsican samples used in the variant discovery. They would also like to thank Julien Marquis and Melanie Dupasquier of the GTF facility for assistance in library preparation and sequencing of the samples used in the study. The authors also thank Milan Malinsky for helpful discussions on analysis and interpretation and Anna Hewett for comments and careful proofreading of the final version. Lastly, the work and manuscript greatly benefited from the insightful comments of two anonymous reviewers.

## Funding

This work was supported by the Swiss National Science Foundation (SNF, grant 310030_215709) to JG.

## Conflicts of interest

The authors declare no conflicts of interest.

## Author contributions

AT and JG devised the project. AR and NP provided the samples. A-LD, CS, and APM carried out the DNA extraction and library preparation. AT, EL, and TC generated and filtered the variant data set. AT performed analyses and wrote the manuscript with input from all coauthors.

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

*Editor: S. Wright*