## [Peer Review File · Genetics]

The recombination landscape of the barn owl, from families to populations

Alexandros Topaloudis, Eléonore Lavanchy, Tristan Cumer, Anne-Lyse Ducrest, Celine Simon, Ana Paula Machado, Nika Paposhvili, Alexandre Roulin, and Jérôme Goudet

NOTE: The reviews and decision letters are unedited and appear as submitted by the reviewers.

In extremely rare instances and as determined by a Senior Editor or the EIC, portions of a review may be redacted. If a review is signed, the reviewer has agreed to no longer remain anonymous.

The review history appears in chronological order.

Review Timeline:

Submission Date:	2024-04-10
Editorial Decision:	2024-05-22
Resubmission Received:	2024-07-26
Editorial Decision:	2024-09-06
Resubmission Received:	2024-10-23
Accepted:	2024-11-01

May 22, 2024

GENETICS-2024-306992

The recombination landscape of the barn owl, from families to populations

Dear Dr. Topaloudis:

Two experts in the field have reviewed your manuscript, and I have read it as well. Both reviewers and I found the study to be well-written and interesting, and provides important new results advancing our understanding of recombination rate evolution and variation. While your manuscript is not currently acceptable for publication in GENETICS, we would welcome a substantially revised manuscript. Both reviewers have comments and concerns to be addressed in a revised manuscript. You can read their reviews at the end of this email.

The reviewers highlight a number of important suggestions and concerns that would need to be addressed in a revision. You will see they are very synergistic and I think all points are important. I would particularly highlight:

- 1) Given the power of having a genome assembly, additional analyses that associate sex-specific recombination rates with gene and repeat density, centromere locations (candidate centromere locations can be identified using software such as Repeatobserver) , could generate important new insights into the correlates of recombination rate variation, and the factors governing heterogeneity in heterochiasmy.
- 2) There are important concerns about the power of the LD-based method to accurately detect differences in hotspots, and the assumptions made in Pyrho. How robust are the inferences of differences in hotspots across populations, particularly to demographic uncertainties and mutation rate uncertainties?
- 3) Both reviewers highlight the importance of better motivating the introduction with broad questions for a GENETICS readership.

I recognize that 1 and 2 likely require fairly extensive reanalysis, but we feel this would be essential for reconsideration at GENETICS.

We look forward to receiving your revised manuscript. Please let the editorial office know approximately how long you expect to need for revisions.

Upon resubmission, please include:

1. A clean version of your manuscript;
2. A marked version of your manuscript in which you highlight significant revisions carried out in response to the major points raised by the editor/reviewers (track changes is acceptable if preferred);
3. A detailed response to the editor's/reviewers' feedback and to the concerns listed above. Please reference line numbers in this response to aid the editor and reviewers.

Your paper will likely be sent back out for review.

Additionally, please ensure that your resubmission is formatted for GENETICS
<https://academic.oup.com/genetics/pages/general-instructions>

Follow this link to submit the revised manuscript: Link Not Available

Sincerely,

Stephen Wright
Associate Editor
GENETICS

Approved by:
David Begun
Senior Editor
GENETICS

Reviewer #2 (Comments for the Authors (Required)):

This manuscript by Topaloudis et al investigates the landscape of recombination in the barn owl *Tyto alba*. They using a combined approach of linkage mapping and LD-based mapping to gain a full picture of landscape variation within this species. This work provides novel insights into various correlates of variation at different scales, such as genome features, chromosome size, sex, and populations. Overall, I think this is an important and interesting contribution to the literature and I appreciate that an impressive amount of analysis has been done. Nevertheless, I have some questions and comments about the approaches and also the overall narrative/framing of the study. These are as follows:

1. The authors find that a large number of chromosomes have a linkage map length of less than 50cM, despite having a high markers density and apparently excellent coverage of the genome. However, with obligate crossing over, there is an expectation of a minimum map length of 50cM. Can the authors comment on why many chromosomes have a lower linkage map length than expected? The authors touch upon it briefly in lines 489 - 496, but it is so pervasive in this particular case that I think it deserved further investigation into e.g. the coverage of markers across the genome, explicit discussion of which biological phenomena would lead to this observation, or biases in their analysis where this may have arisen.
 - a. Could the pruning of markers at the ends of chromosomes been too conservative? The blob of markers at the bottom of Figure 2A - are they more concentrated at chromosome ends e.g. like in the Zhang et al (2023) study?
 - b. I understand that the authors wanted to be sure of their identified putative linkage groups, as outlined by their choice of LOD score threshold. However, I wonder if it was a good choice to (as I understand) do a de novo marker ordering exercise when there is an enormous number of SNPs and a relatively small number of meiotic crossovers likely to be in the dataset. However, I also appreciate that the authors want to preserve the high resolution of the recombination landscape. Did the authors try to order purely based on the genome-predicted order and did this (or would this) give anything more sensible?
2. The section on heterochiasmy is rather underdeveloped. It was not clear to me how the distribution of the crossovers differs between the sexes based on Figure 1C, although I agree that it is clear based on Figure S6. For this part of the analysis, was there any consideration of the centromere positions, or if the chromosomes were metacentric or acro/telocentric? A quick web search of the barn owl karyotype seems to show a mix of both types - could this information be used in Figure 1C and related analyses (e.g. Figure 3E) to get a better understanding of the broad-scale landscape variation, rather than just relative to the chromosome ends (which could be centromeres or telomeres)? I also wondered if the patterns were different based on chromosome size e.g. macro vs microchromosomes (as hinted by Figure 1D).
3. The intra-chromosomal shuffling is a nice addition, but there appear to be no methods provided to on how this was estimated. Also, was this based on the linkage map distances, or individual meioses?
4. In Figure S6, there seems to be an interesting observation of large regions of completely suppressed recombination. Can the authors comment on why this is? Does this correspond to particular genomic features or phenomena e.g. pericentromeric suppression of recombination and/or large structural variants such as inversions? Would it be appropriate to repeat some of the chromosome level statistics (e.g. those described in Figure 2) but excluding these large suppressed regions? At the very least, this should be discussed in more detail as it looks fairly localised. Following on from that, in the illustrative comparison in Figure 4B, this seems to overlap one of these regions on LG32 in Figure S6. Could this be a mechanism to explain population differences in the local rates?
5. After reading the discussion of the hotspot analysis, I would like to see more justification about why Pyrho was chosen for this analysis. As far as I know (based on reading their github, albeit a while ago) that their scaling of rho works well in human genomes, but might hadn't been tested extensively in species with different mutation rates (or mutation rate to recombination ratios). Did the authors consider other potential approaches? Are there differences in hotspot conservation in the genome (e.g. are problems elevated for smaller chromosomes)? Is it sensitive to the choice of N_e (which seemed very large based on Figure S2)? The discussion also states that "researchers can rigorously validate inferred hotspots before drawing conclusions about their evolution of stability" but I'm not sure that this is what the authors have done in this case.

Other comments:

INTRODUCTION

The article summary falls a bit flat - as I say below, this study is a rare example of investigating both contemporary and historical variation in recombination different scales (genome, chromosome, population, sex), and highlight the main findings of the study.

Lines 52 - 59: For the last sentence, the authors could use the terminology of "obligate" crossover and perhaps define

aneuploidy.

Lines 67 - 84: The narrative/rationale for the study here is framed slightly awkwardly e.g. that recombination needs to be understood to interpret various processes, and that it is very difficult to measure, that historical estimates are solving a "problem". I completely agree with these statements, but I also think it directs attention to methods and problems, rather than understanding recombination itself. A real strength of this study is that it investigates recombination at different timescales (contemporary and historical), and this could be brought more to the forefront. Also, perhaps rather than "laborious", the authors could say that it is "challenging" and hit home that using only one approach alone will not give the full picture?

Line 68-69: Is this always the case? Given the advances in genomics since 1992, the authors should cite more modern studies here (e.g. Campos JL et al 2014. *Mol. Biol. Evol.* 31:1010-1028) and perhaps check/give more evidence that this is a universal phenomenon.

Lines 68-71: there are several other phenomena that are also affected by recombination and recombination landscapes, e.g. patterns of adaptive introgression, speciation. A suggestion - perhaps this section could be framed in terms of (a) processes affected by recombination, and (b) interpretation affected by recombination? These are two distinct angles.

Lines 74 - 77: Given that there is a lot of discussion dedicated to LD-based methods, perhaps give a bit more information for the less-familiar reader here e.g. what information that linkage mapping is using and what is required (co-inheritance of alleles).

Line 81: in fairness, there is quite a lot of linkage mapping data out there, including non-model systems (see dataset in Stapley et al 2017) but it is also very fair to say that not much is known about high-resolution recombination landscapes as all those studies have fairly low marker density, and will capture very few crossovers per meiosis. This could be a good point to emphasise.

Lines 95 - 102: This part is a bit awkward. Perhaps this could be better framed in terms of:

- Recombination often occurs in hotspots
- These hotspots are either mediated by the rapidly evolving protein PRDM9 (mammals, some lizards, teleost fish) with high turnover, or...
- The (ancestral) hotspot mode is to have hotspots around functional elements, which are likely stable over millions of years (e.g. bird, dogs)
- Or have no hotspots (*C. elegans*)

Lines 140 - 142: I don't agree with this statement - I think that differences in the rate and landscape can be reconciled as they are not the same thing. Species could have the same recombination hotspots, but could have differences in the rate due to e.g. reduced crossover interference allowing more crossovers to be placed, or due to some sort of dosage mechanism of the proteins that confer higher rates of crossover designation (e.g. like RNF212 in mammals <https://pubmed.ncbi.nlm.nih.gov/23396135/>).

RESULTS

Check that the Supplementary data is referenced in the correct order.

Lines 170 - 171: For coverage, put 16X, range = 8X to 43X

Lines 183: Barn owl genome assembly? Version?

Lines 182 - 184: It is weird that there is a super scaffold, but that here it is two distinct linkage groups. Why has this happened? Did the authors try to make a linkage map with markers against the whole scaffold and find a 50cM gap? (You don't have to do this, I am honestly just curious!!)

Lines 184 - 185: Of those 39 linkage groups, how much of the genome do they cover (in terms of the 38 scaffolds they map to)?

A few requests for the linkage mapping section: how many meiotic crossovers are captured in this dataset? What is the marker distribution resolution (e.g. mean/median intra-marker distance)?

Lines 191 - 192: This could be confusing to someone who is not an expert - a crossover will increase an individual's linkage map by 50cM. This could be framed in that an obligate crossover will result in a minimum length of 50cM.

Figure 1: The little key on Figs 1A and 1B could be made slightly bigger. The sample size stated how many individual, but how many meioses are actually captured here?

Figure S6: There is no sex key here.

Figure 2C: it could be made more explicit here that this is the cM recombination (x) against the LD-estimated recombination (y)

Lines 304 - 306: I kind of agree, but it could also be driven by broad-scale features of the genome. One could argue that Figure 2B is more supportive of the existence of hotspots. (I now see this is discussed in the discussion, so maybe say Figures 2B and C)

Lines 320 - 322: Perhaps briefly mention why higher GC content supports them being hotspots?

DISCUSSION

Lines 400 - 402: I would exert caution here, as there this is a correlation with no explicit test of causation here.

Lines 410 - 435: This narrative comes a bit out of nowhere and is not directly relevant to the results that were just presented. Perhaps the authors could mention in the methods that the linkage mapping was also used to aid with the reference assembly? Otherwise, perhaps this section could be pared down and/or moved to later in the discussion.

Lines 452 - 454: by "linkage group level" do the authors mean the overall length of the chromosome? I agree with that, and perhaps make this clearer.

Sardell and Kirkpatrick is 2020 I think!

Lines 476 - 479: See my major comment about this above - it looks like there is around 1CO per chromosome (= 50cM) and that you may be missing some crossovers.

Lines 575 - 577: What did the Kawakami and Singhal papers find in terms of the proportion of shared hotspots? How do your results compare to other studies?

METHODS

Lines 637 - 669: Put in the reference genome information and version somewhere in this section

Lines 670 - 685: Put in packages and R versions here

Lines 680: What is SNPRelate, and the k1 and k2 statistics?

Lines 710: What did the home-made script do? The analysis should still be reproducible in the methods.

Reviewer #3 (Comments for the Authors (Required)):

This interesting paper uses two complementary approaches to quantify recombination rate variation in barn owls. The data presented are compelling, but the paper could be improved in three ways: the motivation and background could be presented better, there are important missed opportunities for a straightforward analyses, and the sample sizes and details of the description of the methods leave me with some hesitation.

On the motivation and background, both the introduction and the discussion frame the importance and diversity of recombination rate variation and heterochiasmy narrowly. Both sections could be improved by a more global view of recombination in eukaryotes, situating barn owls in the context of birds, vertebrates generally, and animals and plants more broadly. For example, the possible effects of ZW as opposed to XY sex determination on global recombination are not mentioned anywhere. In general, the breadth of background, particularly in plants, needs improvement - even hermaphroditic plants exhibit sex differences in recombination, and more attention could be paid to the taxonomic scope of the phenomenon (as is done in Sardell and Kirkpatrick 2020, which is cited). Currently, the description of recombination rate variation very much frames PRDM9-mediated recombination as the default, but many vertebrate taxa - and all plant, insect, and fungal taxa as far as I know - lack this. In a paper about birds, why not treat mammals as the exception, not the rule? Also, are there any data from other ZW species, such as Lepidoptera?

On missed opportunities: the genome assembly cited includes both an annotation and the identification of the sex chromosomes, yet the analyses presented here don't seem to use either. Describing how recombination patterns differ between the autosomes and the ZW should be possible and interesting: the pedigrees should confirm the identity of the sex chromosomes, and they could then be compared to the autosomes to see if the recombination rates are reduced as expected and identify the pseudoautosomal region. Similarly, since gene density is often correlated with recombination rates, the annotation would allow

the comparison of recombination rates in gene-rich and gene-poor regions. This is a more ambitious set of analyses and I can see the logic of not adding them or saving them for a different paper, but I think the omission of the sex chromosome is a real concern. More generally, when comparing the linkage map to the assembly, the information about the karyotypes and the dot chromosomes should come much earlier.

Finally, the methods: the evidence of hotspots is really interesting, but as the discussion concedes, estimation of recombination is quite challenging. Some additional information in the results about the number of SNPs supporting hotspots and their depth would reassure a reader that these are real hotspots and not genotyping errors (which can, of course, expand maps and create spurious recombination event artifacts).

And a minor comment: the figure captions could be improved. While the writing is in general clear, in the figure captions the descriptions of symbols and abbreviations are often wordy and awkward.

Specific comments:

86 - reduced representation genotyping can also do this in larger sample sizes. Fewer sites sampled, but more recombination events.

93 - include more plant background. They are a major group of eukaryotes with recombination rate variation. In general, be more systematic about this background.

116 - again, reduced representation genotyping has a role here.

124ff - the importance of better understanding this in birds seems undersold.

140 - this is an example of treating PRDM9 as the default when that's perhaps not the best approach.

185ff - explain the dot chromosomes here, and perhaps find the sex chromosomes.

209 - in other words, male recombination is less evenly distributed?

237 - shouldn't there be hyphens in fine(r)-scale?

240 - cite pyrho

295ff - this seems like a relevant place to use the annotation - in particular, where are recombination events relative to exons and introns and how does that relate to more general patterns?

343ff - with these small sample sizes, maybe offer more detail about how the large number of SNPs in WGS adds power? This is still just not capturing that many haplotypes, but should be capturing each one in enough resolution to find historical recombination events - but perhaps unpack that? This would also be the place to add details about the depth, coverage, and number of separate SNPs supporting these local hotspots, just to assure us that they're not genotyping errors.

391 - "we know little about the recombination landscape of most species" doesn't actually seem like very powerful motivation. Motivate this better in the introduction and here - why does having a more comprehensive understanding of recombination landscapes from more species help us understand how recombination evolves? What are the constraints and evolutionary mechanisms in play?

516 - more plant background might help the reader determine whether or not plants are in fact relevant eukaryotes

533 - gene density might also be an important confounding factor, and could be addressed with the annotation

536ff - this is where the "treating mammals as the default" issue seems strongest

558ff - holocentric species come a little out of left field, and could be introduced earlier in a broader overview of recombination rate variation in eukaryotes, which is genuinely fascinating and puzzling

561 - "hotspots ... which were mostly located in regions of low recombination" - this is why it would be good to add that extra verification. Also, are these areas low in genes or high in repetitive sequence? That makes error more likely.

568 - why is there so much about PRDM9 in this paper when birds - as well as plants, insects, lots of fish, crocodiles, and many other eukaryotes - don't even have it? It could be considered briefly and set aside.

Associate Editor Comments:

PS. for all lines references here we refer to the file with the tracked changes.

Reviewer #2 (Comments for the Authors (Required)):

This manuscript by Topaloudis et al investigates the landscape of recombination in the barn owl Tyto alba. They using a combined approach of linkage mapping and LD-based mapping to gain a full picture of landscape variation within this species. This work provides novel insights into various correlates of variation at different scales, such as genome features, chromosome size, sex, and populations. Overall, I think this is an important and interesting contribution to the literature and I appreciate that an impressive amount of analysis has been done. Nevertheless, I have some questions and comments about the approaches and also the overall narrative/framing of the study. These are as follows:

1. The authors find that a large number of chromosomes have a linkage map length of less than 50cM, despite having a high markers density and apparently excellent coverage of the genome. However, with obligate crossing over, there is an expectation of a minimum map length of 50cM. Can the authors comment on why many chromosomes have a lower linkage map length than expected? The authors touch upon it briefly in lines 489 - 496, but it is so pervasive in this particular case that I think it deserved further investigation into e.g. the coverage of markers across the genome, explicit discussion of which biological phenomena would lead to this observation, or biases in their analysis where this may have arisen.

a. Could the pruning of markers at the ends of chromosomes been too conservative? The blob of markers at the bottom of Figure 2A - are they more concentrated at chromosome ends e.g. like in the Zhang et al (2023) study?

b. I understand that the authors wanted to be sure of their identified putative linkage groups, as outlined by their choice of LOD score threshold. However, I wonder if it was a good choice to (as I understand) do a de novo marker ordering exercise when there is an enormous number of SNPs and a relatively small number of meiotic crossovers likely to be in the dataset. However, I also appreciate that the authors want to preserve the high resolution of the recombination landscape. Did the authors try to order purely based on the genome-predicted order and did this (or would this) give anything more sensible?

The reviewer is correct to point out that some of our linkage groups have less than 50cM of genetic map length. Initially, we considered this an unavoidable obstacle of dealing with incomplete genome assemblies but indeed further investigation was required.

- a. The pruning of cM jumps at the end of linkage groups could have led to a decrease of the genetic map length. We looked at the total genetic length before and after pruning the end markers and we saw that these linkage groups had less than 50cM even before pruning (Figure S10 in previous version). Therefore the issue occurred before linkage map building. The blob of markers from Figure 2A is actually located in the centre of our linkage groups as glimpsed from Fig S and seen in the histogram below so that**

wouldn't be the issue.

- b. We are extremely thankful to the reviewer for this suggestion, as it has improved the family recombination map and results. Initially, we performed a de-novo ordering of SNPs and then applied a GAM smoothing model to verify the genetic map ordering of our assembly. While this part is useful in identifying linkage groups, it could lead to a misordering in certain regions due to the small number of meioses in the dataset. Therefore, we repeated linkage mapping using a less stringently (only MAF < 10% and missingness > 10%) filtered set of SNPs. Then we ran “SeparateChromosomes” using a LOD score cutoff of 5 (LOD5 dataset) and built a genetic map on the physical order of those markers by modifying the “OrderMarkers” step of LepMAP3. The motivation was to include a much larger set of markers to make sure we were not missing crossover events due to aggressive filtering in the previous order. After ordering, we still pruned markers with jumps larger than 2cM at the end of the linkage groups and produced a new set of linkage maps that is now used for the results of the study.

Since this update changed the whole dataset used, it is documented in Methods (lines 1004-1014) and Results 250-255. A comparison of the three datasets (de-novo ordering at LOD15 + physical ordering at LOD5) is given in Figure S10 in file S1. Notably, now most linkage groups have a sex-averaged genetic length of more than 50cM, except LG13 and LG31 with 40 and 34.8cM respectively (LG18 has a sex-averaged length of 49.2cM). As seen from Figure S10, the gain of genetic length comes from including SNPs closer to the end of the assembled sequence (ex LG5 or LG21). Lastly, the genetic length is similar to pre-pruning of the end markers so the ‘missing cM issue’ must extend to the input data and thus to the assembly stage. This is now discussed briefly in lines 683-692.

This approach also allowed us to quantify the full genetic map of the merged linkage group (LG20, which includes Super-Scaffold_3 and Super-Scaffold_49) and the Z chromosome (now LG40, which includes Super-Scaffold_13 and Super-Scaffold_42).

2. The section on heterochiasmy is rather underdeveloped. It was not clear to me how the distribution of the crossovers differs between the sexes based on Figure 1C, although I agree that it is clear based on Figure S6. For this part of the analysis, was there any consideration of the centromere positions, or if the chromosomes were metacentric or acro/telocentric? A quick web

search of the barn owl karyotype seems to show a mix of both types - could this information be used in Figure 1C and related analyses (e.g. Figure 3E) to get a better understanding of the broad-scale landscape variation, rather than just relative to the chromosome ends (which could be centromeres or telomeres)? I also wondered if the patterns were different based on chromosome size e.g. macro vs microchromosomes (as hinted by Figure 1D).

We are not sure how else to represent the difference in the position of crossovers between males and females. We thank the reviewer for the comment, however we believe that Fig 1C represents the sex differences because it describes the placement of crossovers across sexes and chromosomes. We have also added a supplementary figure (S2 in file S1) with the actual crossover locations along the length of the linkage groups. Thank you for the useful suggestion to use a centromere annotation. We used two software to annotate their positions, RepeatObserver and TRASH, both of which use the repeat content along the autosomes to look for putative centromere positions. This is now documented in Supplementary Text S2 and the final annotation is shown in Figure S4 in file S1 which is included here:

Caption: Local polynomial regression for male and female relative recombination rates (divided by the mean of the rate in the linkage group) in 1Mb windows for all linkage groups with an annotated centromere. The putative centromere is at position 0 of the x-axis. Number below the x-axis is the length of the linkage group. Top left plot is the plot of all linkage groups taken together.

Interestingly, we believe that we still can not be confident in the centromere annotation of our assembly. Karyotypic results of our species show acro- or telocentric autosomes and a metacentric Z chromosome (Belterman and De Boer 1984; Rebholz *et al.* 1993). We used this information to disregard centromere annotations in non-distal regions of our chromosomes (see Supp. Text S2) but we can not be sure if the end annotated corresponds to a centromere or a telomere since both harbour low-diversity, high-density repeats. In fact, quantification of recombination rates in putative centromere positions in figure S4 above often does not show recombination suppression as would be expected, supporting the possibility that the annotated end is indeed a telomere - note however that often the opposite end shows recombination suppression. This is now briefly discussed in results 290-292 and discussion 627-632. The methodology is summarised in methods 1057-1060 and Supplementary Text S2.

3. *The intra-chromosomal shuffling is a nice addition, but there appear to be no methods provided to on how this was estimated. Also, was this based on the linkage map distances, or individual meioses?*

While we briefly mentioned calculating r_{intra} at lines 763-764 and we included the script used on the github page under the name 4.8.rintra.r, we understand that the description is poor and the script not easily identifiable.

In the new version, because we use the physical maps we tweaked how we calculate R-intra. It is now mentioned in methods 1064-1069 and the accompanying script is R.rintra.r.

4. *In Figure S6, there seems to be an interesting observation of large regions of completely suppressed recombination. Can the authors comment on why this is? Does this correspond to particular genomic features or phenomena e.g. pericentromeric suppression of recombination and/or large structural variants such as inversions? Would it be appropriate to repeat some of the chromosome level statistics (e.g. those described in Figure 2) but excluding these large suppressed regions? At the very least, this should be discussed in more detail as it looks fairly localised. Following on from that, in the illustrative comparison in Figure 4B, this seems to overlap one of these regions on LG32 in Figure S6. Could this be a mechanism to explain population differences in the local rates?*

Indeed we identify clear ‘deserts’ of recombination on multiple chromosomes. While some distal deserts might be due to a pericentromeric suppression of recombination, most of these regions are found in the middle of the chromosomes and are probably not caused by centromeres (since all chromosomes are expected to be acrocentric/ telocentric).

We tested if these regions are due to our application of a mappability mask (i.e. removing all variants from 10% of the genome) but the number of SNPs in these regions is not abnormal and the proportion masked on these windows is not high. Therefore, to our knowledge these regions show a true signal of reduced recombination. We did not test for the presence of polymorphic inversions though this could be a possible explanation.

This is now mentioned more explicitly in the manuscript, for example in lines 541-542, 707-708.

5. After reading the discussion of the hotspot analysis, I would like to see more justification about why Pyrho was chosen for this analysis. As far as I know (based on reading their github, albeit a while ago) that their scaling of rho works well in human genomes, but might hadn't been tested extensively in species with different mutation rates (or mutation rate to recombination ratios). Did the authors consider other potential approaches? Are there differences in hotspot conservation in the genome (e.g. are problems elevated for smaller chromosomes)? Is it sensitive to the choice of N_e (which seemed very large based on Figure S2)? The discussion also states that "researchers can rigorously validate inferred hotspots before drawing conclusions about their evolution of stability" but I'm not sure that this is what the authors have done in this case.

We chose Pyrho because of its ease of implementation, its incorporation of demographic history and the use of non-phased data. We believe that the rho scaling would be problematic regardless. Other software like LDhat or LDhelmet would provide no rho scaling and instead report the ρ / bp rates. Here pyrho attempts to transform this to cM / Mb. Uncertainties in demographic history and the mutation rate will affect this and lead to erroneous estimates of recombination rates. The saving grace is that relative rates would be correctly identified and then a simple scaling would account for this error. In this case, scaling the result of pyrho to be the same as the genetic length inferred from LepMAP3 accounts for the absolute value difference. For that reason, we did not consider other approaches.

Pyrho has been compared with LDhelmet in the original publication and also independently in the Supplementary Material S5.4 of Hoge et al., 2024. In the latter, the authors found some spurious signals in pyrho hotspots, specifically a correlation between elevated allele frequencies and π with recombination rates and conclude: "While these relationships could be real, a concern is that by Pyrho sometimes infers an elevation in the population recombination rate when there is a deep gene genealogy rather than elevated recombination rates". To our knowledge there is no other evidence of this and this is an expected outcome of recombination in a lot of species, and as illustrated in the updated Fig3 also in the barn owl.

However, in the updated version of the manuscript we apply further filters on the pyrho SNP dataset and only compare hotspots that are found in intermediate recombination rates (> 1 & < 10 cM/Mb) where recombination inference is most powerful (Singhal et al., 2015). We also keep hotspot windows with no more than half the window annotated as a repeat. Furthermore, we tested the heterozygosity of SNPs in hotspot windows and found it matching a random set of non-hotspot windows. Lastly, by using 5 subsets of Swiss individuals with $n=13$ we can get an idea what hotspot sharing comes from statistical noise in inference and what from true underlying differences.

Other comments:

The article summary falls a bit flat - as I say below, this study is a rare example of investigating both contemporary and historical variation in recombination different scales (genome, chromosome, population, sex), and highlight the main findings of the study.

We rewrote the article summary.

Lines 52 - 59: For the last sentence, the authors could use the terminology of "obligate" crossover and perhaps define aneuploidy.

The end of the paragraph now does this (lines 64-70)

Lines 67 - 84: The narrative/rationale for the study here is framed slightly awkwardly e.g. that recombination needs to be understood to interpret various processes, and that it is very difficult to measure, that historical estimates are solving a "problem". I completely agree with these statements, but I also think it directs attention to methods and problems, rather than understanding recombination itself. A real strength of this study is that it investigates recombination at different timescales (contemporary and historical), and this could be brought more to the forefront. Also, perhaps rather than "laborious", the authors could say that it is "challenging" and hit home that using only one approach alone will not give the full picture?

We have reformatted the relevant part of the introduction following the reviewer's comments. (lines 78-102)

Line 68-69: Is this always the case? Given the advances in genomics since 1992, the authors should cite more modern studies here (e.g. Campos JL et al 2014. Mol. Biol. Evol. 31:1010-1028) and perhaps check/give more evidence that this is a universal phenomenon.

We have added more recent citations and mentioned the uncertainty across species. (lines 87-90)

Lines 68-71: there are several other phenomena that are also affected by recombination and recombination landscapes, e.g. patterns of adaptive introgression, speciation. A suggestion - perhaps this section could be framed in terms of (a) processes affected by recombination, and (b) interpretation affected by recombination? These are two distinct angles.

We have added a clearer distinction in these paragraphs (lines 78-102).

Lines 74 - 77: Given that there is a lot of discussion dedicated to LD-based methods, perhaps give a bit more information for the less-familiar reader here e.g. what information that linkage mapping is using and what is required (co-inheritance of alleles).

We have added the short description suggested (lines 105-107).

Line 81: in fairness, there is quite a lot of linkage mapping data out there, including non-model systems (see dataset in Stapley et al 2017) but it is also very fair to say that not much is known about high-resolution recombination landscapes as all those studies have fairly low marker density, and will capture very few crossovers per meiosis. This could be a good point to emphasise.

We have highlighted the lack of high resolution datasets instead of claiming the lack of data. (lines 112 - 114)

Lines 95 - 102: This part is a bit awkward. Perhaps this could be better framed in terms of:

- Recombination often occurs in hotspots
- These hotspots are either mediated by the rapidly evolving protein PRDM9 (mammals, some lizards, teleost fish) with high turnover, or...
- The (ancestral) hotspot mode is to have hotspots around functional elements, which are likely stable over millions of years (e.g. bird, dogs)
- Or have no hotspots (*C. elegans*)

We thank you for the suggestion. The paragraph has been modified accordingly (133 - 143).

Lines 140 - 142: I don't agree with this statement - I think that differences in the rate and landscape can be reconciled as they are not the same thing. Species could have the same recombination hotspots, but could have differences in the rate due to e.g. reduced crossover interference allowing more crossovers to be placed, or due to some sort of dosage mechanism of the proteins that confer higher rates of crossover designation (e.g. like RNF212 in mammals <https://pubmed.ncbi.nlm.nih.gov/23396135/>).

The reviewer is right and this claim has been removed.

Check that the Supplementary data is referenced in the correct order.

We have been through supplementary references to verify that.

Lines 170 - 171: For coverage, put 16X, range = 8X to 43X

Changed as suggested. (I230)

Lines 183: Barn owl genome assembly? Version?

Added version. (I239)

Lines 182 - 184: It is weird that there is a super scaffold, but that here it is two distinct linkage groups. Why has this happened? Did the authors try to make a linkage map with markers against the whole scaffold and find a 50cM gap? (You don't have to do this, I am honestly just curious!!)

This is based on the linkage mapping information on the SeparateChromosomes step. The result split Super-Scaffold_2 into linkage groups 1 and 24. Before, the Super-Scaffold was concatenated with approx. 10'000 N nucleotides based on Bionano information and showed a length similar to the Z chromosome (which would be too big based on karyotypic information). We were thus quite happy to split it according to LepMAP3.

Lines 184 - 185: Of those 39 linkage groups, how much of the genome do they cover (in terms of the 38 scaffolds they map to)?

We have moved this information in that sentence to make it clearer. Numbers now represent the results with the new dataset. (I240-242)

A few requests for the linkage mapping section: how many meiotic crossovers are captured in this dataset? What is the marker distribution resolution (e.g. mean/median intra-marker distance)?

We now include this information. (I240-242, I252-253, Fig 1)

Lines 191 - 192: This could be confusing to someone who is not an expert - a crossover will increase an individual's linkage map by 50cM. This could be framed in that an obligate crossover will result in a minimum length of 50cM.

We have rephrased this. (I261-262)

Figure 1: The little key on Figs 1A and 1B could be made slightly bigger. The sample size stated how many individual, but how many meioses are actually captured here?

Changed key size and added # of meioses observed in pedigree. (Fig 1, main text)

Figure S6: There is no sex key here.

It was only in the last plot but now it is in every plot. (Fig S1 file S1)

Figure 2C: it could be made more explicit here that this is the cM recombination (x) against the LD-estimated recombination (y)

Now reads family estimate and LD-based estimate. (Fig 1, main text)

Lines 304 - 306: I kind of agree, but it could also be driven by broad-scale features of the genome. One could argue that Figure 2B is more supportive of the existence of hotspots. (I now see this is discussed in the discussion, so maybe say Figures 2B and C)

Added both when we mention it. (I499)

Lines 320 - 322: Perhaps briefly mention why higher GC content supports them being hotspots?

Since this part is now removed we only discuss this in a different light (I753-755).

Lines 400 - 402: I would exert caution here, as there this is a correlation with no explicit test of causation here.

You are right to point this out. We have appropriate wording this time around. (I543)

Lines 410 - 435: This narrative comes a bit out of nowhere and is not directly relevant to the results that were just presented. Perhaps the authors could mention in the methods that the linkage mapping was also used to aid with the reference assembly? Otherwise, perhaps this section could be pared down and/or moved to later in the discussion.

We think this is relevant because it is linked to the de-novo ordering of the sequences into linkage groups presented in the first paragraph of the results. For this reason and because we now include the Z chromosome, we have kept this part but shortened it so that it doesn't take up too much of the discussion.

Lines 452 - 454: by "linkage group level" do the authors mean the overall length of the chromosome? I agree with that, and perhaps make this clearer.

Clarified (I587-588).

Sardell and Kirkpatrick is 2020 I think!

Absolutely! Unsure why zotero had this noted down as 2019. Corrected now.

Lines 476 - 479: See my major comment about this above - it looks like there is around 1cM per chromosome (= 50cM) and that you may be missing some crossovers.

Corrected through new dataset.

Lines 575 - 577: What did the Kawakami and Singhal papers find in terms of the proportion of shared hotspots? How do your results compare to other studies?

Number of hotspots inferred will differ between methods, demographics and parameters used (for example as mentioned in Supplementary text in Hoge et al., 2024). We do not believe that absolute numbers would help in this case. Kawakami 2017 has similar proportion of hotspot sharing while Singhal has higher (66% are shared between species). Both (and we now) focused only on relative hotspot sharing.

METHODS

Lines 637 - 669: Put in the reference genome information and version somewhere in this section

We have added the genome assembly information. (I935)

Lines 670 - 685: Put in packages and R versions here

We have corrected this. (I966-975)

Lines 680: What is SNPRelate, and the k_1 and k_2 statistics?

We have added this information (I972-973)

Lines 710: What did the home-made script do? The analysis should still be reproducible in the methods.

Reworded to explain that. (I1016-1018)

Reviewer #3 (Comments for the Authors (Required)):

This interesting paper uses two complementary approaches to quantify recombination rate variation in barn owls. The data presented are compelling, but the paper could be improved in

three ways: the motivation and background could be presented better, there are important missed opportunities for a straightforward analyses, and the sample sizes and details of the description of the methods leave me with some hesitation.

On the motivation and background, both the introduction and the discussion frame the importance and diversity of recombination rate variation and heterochiasmy narrowly. Both sections could be improved by a more global view of recombination in eukaryotes, situating barn owls in the context of birds, vertebrates generally, and animals and plants more broadly. For example, the possible effects of ZW as opposed to XY sex determination on global recombination are not mentioned anywhere. In general, the breadth of background, particularly in plants, needs improvement - even hermaphroditic plants exhibit sex differences in recombination, and more attention could be paid to the taxonomic scope of the phenomenon (as is done in Sardell and Kirkpatrick 2020, which is cited). Currently, the description of recombination rate variation very much frames PRDM9-mediated recombination as the default, but many vertebrate taxa - and all plant, insect, and fungal taxa as far as I know - lack this. In a paper about birds, why not treat mammals as the exception, not the rule? Also, are there any data from other ZW species, such as Lepidoptera?

We thank you for your comments. We have attempted to broaden the scope of our introduction and discussion, to place our study within the global knowledge of other taxonomic groups - including references to the plant literature. We also limited our comparisons to PRMD9+ species which are only discussed briefly in the introduction and discussion and only for hotspot comparisons. We feel like this has greatly improved our writing and gives more weight to our claims and conclusions.

On missed opportunities: the genome assembly cited includes both an annotation and the identification of the sex chromosomes, yet the analyses presented here don't seem to use either. Describing how recombination patterns differ between the autosomes and the ZW should be possible and interesting: the pedigrees should confirm the identity of the sex chromosomes, and they could then be compared to the autosomes to see if the recombination rates are reduced as expected and identify the pseudoautosomal region. Similarly, since gene density is often correlated with recombination rates, the annotation would allow the comparison of recombination rates in gene-rich and gene-poor regions. This is a more ambitious set of analyses and I can see the logic of not adding them or saving them for a different paper, but I think the omission of the sex chromosome is a real concern. More generally, when comparing the linkage map to the assembly, the information about the karyotypes and the dot chromosomes should come much earlier.

Following your suggestion we have now included gene content and the Z chromosome whose genetic map was built using the physical order as suggested by Reviewer 2. Figure 3 was modified to highlight correlates of recombination in the barn owl.

Including the W chromosome could be done through pedigree data from a set of unmapped reads but we feel that this will include further extensive analyses beyond the scope of our study.

Finally, the methods: the evidence of hotspots is really interesting, but as the discussion concedes, estimation of recombination is quite challenging. Some additional information in the results about the number of SNPs supporting hotspots and their depth would reassure a reader

that these are real hotspots and not genotyping errors (which can, of course, expand maps and create spurious recombination event artifacts).

Due to the doubt raised by both reviewers, we sought to validate the hotspots annotated by pyrho. We tested the variant quality, depth, missingness and heterozygosity of the snps placed in 1kb windows belonging to recombination hotspots against an equally-sized sample of non-hotspot windows. We found that hotspots exhibited lower depth and quality and higher missingness than non-hotspot windows. They further showed a noise relationship between heterozygosity and allele frequency. All results can be seen in the figure below:

That showed an issue with the input data and we went back to re-generate our data for pyrho. Previously we filtered out singletons and sites with more than 10% missingness along with standard GATK filters, a mappability mask and an individual depth filter. Now

we include an allele frequency filter (filtering out SNPs with $MAF < 5\%$) and a HWE filter (filtering out SNPs with less than 0.05 p-value in a Fisher's exact test) to rule out the presence of errors (despite throwing out some true positive variants). However, for pyrho, the authors note that they don't know how filtering will affect the results and we feel it is beyond our scope to test that.

Furthermore, as you pointed out in a later comment we tested if hotspots are found mostly in regions of low gene density or high repeat density by looking at the distribution of the 1kb windows that overlaps genes or repeats. While there was no obvious difference in gene density, a fifth of the hotspots were in 1kb windows filled with repeats. Therefore we filtered out any hotspot window with more than 500bp (half a window - arbitrary cutoff) in an annotated repeat. We also restricted the annotated hotspots to regions of intermediate recombination rates ($> 1 \text{ cM/Mb}$ and $< 10 \text{ cM/Mb}$), shown to be better for hotspot identification (Singhal et al., 2015) and focused only on qualitative overlapping between populations. In addition we use 5 subsamples of $n=13$ from Switzerland to see what hotspot overlap is expected through statistical noise of subsampling. In the end we got 1566 hotspot windows with 8171 SNPs against 1566 nonhotspot windows with 7812 SNPs. The new figure is much improved:

While there is still a deficit in variant quality and an increase in missingness, that might be confounded by high GC content in hotspots since we did not match GC content between our hotspot and non-hotspot windows.

And a minor comment: the figure captions could be improved. While the writing is in general clear, in the figure captions the descriptions of symbols and abbreviations are often wordy and awkward.

We thank the reviewer for their suggestion, and have reworked the figure legends appropriately for succinctness and clarity. Please see the figure legends directly in the manuscript.

Specific comments:

86 - reduced representation genotyping can also do this in larger sample sizes. Fewer sites sampled, but more recombination events.

We added that to the sentence although the software addressed here (LDhat, LDhelmet, pyrho) are meant to work with whole genome sequences. (I121).

93 - include more plant background. They are a major group of eukaryotes with recombination rate variation. In general, be more systematic about this background.

We have now added relevant studies in plants in different places in the manuscript. Lastly, we reformatted this paragraph to remove some focus from PRDM9 species. (133-143). We feel this improves the quality of the writing and better frames our study in what is known in other organisms.

116 - again, reduced representation genotyping has a role here.

Sentence has been removed due to major introduction reworking.

124ff - the importance of better understanding this in birds seems undersold.

We agree and have rephrased the gap of knowledge in the bird recombination literature (169ff).

140 - this is an example of treating PRDM9 as the default when that's perhaps not the best approach.

We agree with the reviewer and removed this part of the paragraph.

185ff - explain the dot chromosomes here, and perhaps find the sex chromosomes.

We added a line mentioning the putative microchromosomes. The dot chromosomes in our opinion belong to the discussion since they have not been identified. Sex chromosome has also been added. (I244-250)

209 - in other words, male recombination is less evenly distributed?

Correct! We use this phrasing now as it is clearer. (1289-290)

237 - shouldn't there be hyphens in fine(r)-scale?

Corrected.

240 - cite pyrho

Cited!

295ff - this seems like a relevant place to use the annotation - in particular, where are recombination events relative to exons and introns and how does that relate to more general patterns?

While we did not test for finer scale effects on exons / introns which would be on the sub-kb scale, we have added the correlation with windowed gene content in Figure 3.

343ff - with these small sample sizes, maybe offer more detail about how the large number of SNPs in WGS adds power? This is still just not capturing that many haplotypes, but should be capturing each one in enough resolution to find historical recombination events - but perhaps unpack that? This would also be the place to add details about the depth, coverage, and number of separate SNPs supporting these local hotspots, just to assure us that they're not genotyping errors.

Following your and reviewer 2's comments about pyrho we shifted the focus of figure 3 from hotspots to more broad-scale correlates of recombination. Now 1kb hotspots are discussed in the population comparison. We added information about the number of SNPs. Concerning the quality of hotspot annotation please refer to the major comment.

391 - "we know little about the recombination landscape of most species" doesn't actually seem like very powerful motivation. Motivate this better in the introduction and here - why does having a more comprehensive understanding of recombination landscapes from more species help us understand how recombination evolves? What are the constraints and evolutionary mechanisms in play?

We have changed the phrasing in this sentence to provide stronger motivation (1527-529). The introduction was reworked based on both reviewers' comments to increase focus on recombination instead of it as a solution to a problem. We did maintain parts the old structure but have discussed key points in more detail.

516 - more plant background might help the reader determine whether or not plants are in fact relevant eukaryotes

We now discuss and cite more plant literature throughout the manuscript.

533 - *gene density might also be an important confounding factor, and could be addressed with the annotation*

We discuss this more now and from a different angle (722-758).

536ff - *this is where the "treating mammals as the default" issue seems strongest*

Now only one paragraph in introduction and discussion read about PRDM9 and only concerning hotspots.

558ff - *holocentric species come a little out of left field, and could be introduced earlier in a broader overview of recombination rate variation in eukaryotes, which is genuinely fascinating and puzzling*

We no longer discuss holocentric species but hope we have added more information about recombination in different species.

561 - *"hotspots ... which were mostly located in regions of low recombination" - this is why it would be good to add that extra verification. Also, are these areas low in genes or high in repetitive sequence? That makes error more likely.*

Thank you for this comment which ended up helping us filter down our inferred hotspots. The relevant discussion part has been completely rewritten.

568 - *why is there so much about PRDM9 in this paper when birds - as well as plants, insects, lots of fish, crocodiles, and many other eukaryotes - don't even have it? It could be considered briefly and set aside.*

Discussion has been mostly rewritten, especially regarding PRDM9 since hotspot focus has been shifted.

September 6, 2024

GENETICS-2024-307315

The recombination landscape of the barn owl, from families to populations

Dear Dr. Topaloudis:

Two experts in the field have reviewed your manuscript, and I have read it as well. I am pleased to inform you that, with minor revisions, it is potentially suitable for publication in GENETICS. The reviewers have comments and concerns that need to be addressed in a revised manuscript. You can read their reviews at the end of this email.

Thank you for your careful consideration of reviewer comments, the revised version has improved considerably. The reviewers have a number of residual suggestions which should be addressed- I would particularly flag the importance of caution when marking arguments about regions of low recombination without making the point about consecutive windows (reviewer 2), and please have a careful round of editing to sharpen up the writing, make firmer concluding sentences and placing the work in the broader context (reviewer 3).

We look forward to receiving your revised manuscript. Please let the editorial office know approximately how long you expect to need for revisions.

Upon resubmission, please include:

1. A clean version of your manuscript;
2. A marked version of your manuscript in which you highlight significant revisions carried out in response to the major points raised by the editor/reviewers (track changes is acceptable if preferred);
3. A detailed response to the editor's/reviewers' comments and to the concerns listed above. Please reference line numbers in this response to aid the editors.

Additionally, please ensure that your resubmission is formatted for GENETICS.

<https://academic.oup.com/genetics/pages/general-instructions>

Follow this link to submit the revised manuscript: Link Not Available

Sincerely,

Stephen Wright
Associate Editor
GENETICS

Approved by:
David Begun
Senior Editor
GENETICS

Reviewer #1 (Comments for the Authors (Required)):

It was pleasure for me to review the article. The authors adjusted the manuscript to the reviewers' suggestions and it was improved. No more suggestion.

Reviewer #2 (Comments for the Authors (Required)):

Thank you for revising your manuscript based on our comments. The manuscript is much improved from the previous version. The reanalyses have made the results much more convincing. Overall, it's a great paper and I think will be a very useful and

welcome addition to the literature. I don't have any suggestions for reanalysis, but I do have some remaining comments on clarity and interpretation. My line numbers refer to the "Track Changes" document.

ABSTRACT

There are a few word/syntax choices that I would change:

"A dearth of" = "limited"

"Unbiased" = remove, and perhaps state "realised recombination"

"We find that recombination rates are correlated with..." would be less awkward

"We find no conclusive differences in..." recombination landscapes?

SUMMARY

"Differences between" - "Differences in rate and landscape between"?

INTRODUCTION

In the Introduction, PRDM9 is introduced (Page 7) but I think in the updated introduction, it is now not explicitly mentioned that birds lack PRDM9 and have stable hotspots. It would be good to mention that somewhere as it is relevant for your study.

Line 80: "it can be beneficial" might be better than "often"

Line 81: delete "different"

Lines 88-90: I didn't quite get this sentence - isn't genetic diversity also a consequence of biased-gene conversion?

Line 121: "models"

Line 136: species names should be in italics, would also precede those species "with some exceptions, such as [...]"

Line 170: "lagged behind" - this might be a little cheeky to the many groups working on recombination in birds :) maybe this could be toned down. I agree that it is rare to look at both contemporary and historical recombination, but actually this is rare for any species!

Lines 185-195: When comparing map lengths, it's important to consider if they have used different mapping functions, especially with lower numbers of markers. This could be a reason why the map lengths could be quite different. FYI, a new study came out that also shows strong heterochiasmy in the hihi (very male biased!): <https://www.nature.com/articles/s41437-024-00711-3>

RESULTS

I just wanted to comment that all the figures look great!

Lines 248 - briefly define a microchromosome?

Line 281: "heterochiasmy among linkage groups at the chromosome scale"

Figure 1: I like the little stats box - could also add the number of COs?

Line 323: Not sure of the journal style, but I think it's good to put the version numbers of softwares in the results too (given that they come before the methods here)

Line 332 and throughout: I think you have to be careful about this statement about regions of low recombination in the linkage mapping. Low recombination regions are to be expected as there isn't a large number of crossovers in the dataset in the first place - perhaps a mean of 4COs per window per sex, meaning that loads of windows will have 0 COs just by chance. Looking at Figure S2, I see that a lot of these windows are adjacent, so you could incorporate this observation in your arguments about low recombination regions (I'm more convinced by long stretches, for example).

Line 355: "leading to" should probably be "associated with"

DISCUSSION

Line 539: Just linking back to what I said above - these statements are a little risky without a bit more justification for "suppressed" recombination.

Lines 574 - 579: This is just a comment, but if the authors know - what do these dot chromosomes do during meiosis? Do they recombine as normal? Does anyone know?

Line 593: If I recall correctly, the great reed warbler had extreme telomeric recombination in males? It might be worth double checking this paper to make sure these were or were not different between the sexes.

Line 595: "dominated" isn't quite the right word - perhaps better to say that there was much higher recombination in females? Also going to flag again the hihi paper with male biased recombination :)

Line 600: "fine-scale information on sex differences"

Line 620: Why is haploid selection more relevant in plants? Maybe can say that there is no female haploid selection (if that's true) - I think this statement is slightly underdeveloped (although there is already a lot of detail in this section!)

Line 667: the sparrow paper will have a value for this chromosome as well (I don't know which chromosome it is!)

Line 681: "were likely not inferred"

Line 681 - 682: This sentence is a bit awkward, perhaps reword to say that the chromosomes that are less than 50cM were probably missing markers?

Line 690: yes - I think this sentence needs a reference for this statement though.

Lines 706 - 710: these are good points. I think these papers are more plant-centric so might be good to qualify that this may also hold for other species.

Lines 712 - 713: maybe due to structural variants? I know that Ulrich Knief had a paper on inversions in zebrafishes that could be a good reference for that point.

Line 733: comma after "In fact"

Line 741: "a few"

Line 781: "inspired by"

Line 786: "confounded"

Line 794 - 795: Is this the case? I feel that it is ore that comparisons have been within PRDM9 or within non-PRDM9. That is kind of what the following sentences discuss.

Line 807 - 808: as you don't develop this idea, perhaps the sentence can begin with a "yet".

Line 879: no comma after "thus".

METHODS

Caveating this section with - I am one of those people that read the methods before the results...!

Throughout - please provide ALL software version numbers, even if they were in the Results, and put all functions in italics. Also check the case (e.g. PLINK, Plink or plink?) of different softwares. There is a bit of a lack of consistency in these regards throughout the methods.

Lines 911 - 918: Put the sequencing coverage information here too

Line 962 - 963: what does it mean to resolve first and second degree links?

Line 976: "Mendelian"

Line 989: I haven't seen the use of the word "thrice" since I read Enid Blyton books, so maybe better to put "three times"

Line 993: "After confirming"

Line 1005: justification of 2cM?

Line 1023: snowy owl (Latin name)?

Line 1032: Where is the Moran approximation implemented? E.g. software?

Line 1053: It has not been included at all how to calculate the intra-chromosomal allelic shuffling! It is in the Veller paper, also sparrows, Cathrine Brekke's recent papers, etc. - please put the equation and be clearer how this was done.

DATA AVAILABILITY

Although a link to the code is signposted, there isn't an explicit data availability statement, please include one.

Reviewer #3 (Comments for the Authors (Required)):

I was pleased to see this manuscript again. The authors have improved it considerably, and it represents a useful and engaging addition to the scientific literature. It would benefit from a few more slight tweaks to better showcase its importance and novelty, and to clarify some methods, but is very close to ready.

General comments on substance:

The research presented here advances the scientific conversation in two directions: first, it improves the understanding of recombination hotspots in species that, like the majority of eukaryotes, lack PRDM9, and second, it expands the taxonomic breadth of (particularly birds) with recombination landscapes characterized, representing the first owl genome and only the third order of birds to be studied in this way.

Given that these are important and ongoing scientific conversations, the introduction could move faster to the importance of hotspots and recombination landscapes and condense the more general discussion of the importance of recombination and the methods used to infer it in favor of setting up the questions this work addresses.

General comments on writing:

The writing is comprehensible but has some cumbersome moments. I've noted some in the specific comments below, but consider giving this a sharp copy edit. Also there are some typos and issues with the citation formatting, so make sure to clean those up.

Several sections, particularly in the discussion and conclusion, have a weak concluding sentence. For example, "Picking apart the magnitude and direction of these forces can help us understand what drives and what is driven by recombination rate variation in different species" is accurate, but doesn't add much insight - it just asserts that science can be done. In general, try to pep these up and perhaps mention some specific follow-up work (particularly the question of adaptive interspecific differences in recombination landscape and hotspot placement - that is interesting!).

Data availability:

The statement looks fine but the XXXX's need to be filled in, of course.

Specific comments:

Abstract:

The sentence featuring "albeit depending" is awkward.
"We find correlations of ..." - why not "gene content etc. correlated with"?

Article summary:

Generally, a place for prose improvement.

Introduction:

I disagree that the inference of recombination rate is overlooked in non-model species. It's a major area of research right now. Perhaps it used to be neglected or is more challenging, but it isn't overlooked now.

The first three paragraphs could be condensed into a single paragraph to get to the novelty sooner. "Thus, to advance our understanding of recombination and distinguish it from other forces that shape the genome, we need to be able to quantify it and understand the sources of its variation in different species and under different evolutionary backgrounds" seems like a thesis statement that could steer that condensed paragraph.

Does the (limited?) scientific understanding of non-PRDM9 mechanisms of recombination hotspots motivate this research?

It's really interesting that the patterns of heterochiasmy are all over the place in birds! That in itself is really interesting and motivating and could come earlier. It's a great reason to research bird recombination.

Methods:

Comment more on these low-mappability regions. Are they annotated? If so, are they repetitive sequence? Where are they located? They're probably not important but add more justification for discarding them.

Why downsample using three scaffolds rather than pruned SNPs distributed across the genome? This is a strange choice.

I may have missed it, but did any loci show evidence of transmission ratio distortion? I'd expect there to be some, and indeed one of the cited papers (Coop and Myers 2007) appears to be about how it relates to hotspots. It also relates to some of the theories described in Sardell and Kirkpatrick for landscape dimorphism. Anyway, if the filtering threw out distorted sites - I'm not sure how the bcftools Mendelian plugin works but if it was set to penalize divergence from HWE rather than just purge impossible sites - then an interesting dimension might have been lost.

This is a hard thing because so many things affect recombination and can bias genotyping and inference, but how confident is it possible to be that these hotspots are not by chance?

Maybe define the Gini coefficient briefly, since people may not be familiar with it outside the socioeconomic application? Especially since the relationship between Gini coefficient and length seems to be an effective way to describe these recombination patterns.

Results:

Perhaps report slightly more about the putative centromeres in the text, especially since bird centromeres are poorly characterized and this is therefore a contribution to identifying those sequences? Since many of the chromosomes are acrocentric, just describe in a bit more detail what repeats were and were not observed in the low-recombination ends? Also, it is entirely fair that centromeres are really hard to assemble and annotate - even the human genome only got that done recently - but perhaps given that limitation the phrasing "near chromosome level" or the clarification "chromosome level, though not telomere-to-telomere" is merited.

It's confusing that in figure 2B the three chromosomes are not in fact at the same scale.

Discussion:

Slightly reordering the sentences about the regular vs. dot chromosomes would be much clearer. "The karyotype of the barn owl contains 45 autosomal pairs, including a large Z chromosome, small degraded W chromosome, and six 'dot' microchromosomes." Then we kind of expect the 39 that were actually assembled. Also, how much evidence is there that the dot chromosomes are biologically relevant? Could the discussion of them be compressed?

Some of the context about recombination landscapes, how little is known about heterochiasmy in birds, and the implications of sex differences in landscape could perhaps be set up in the introduction. Similarly, the section describing inter- and intraspecific variation in recombination landscape is really interesting and positions this in the context of broader comparative work, so a hint of it could be used to motivate the study.

For all line numbers, we refer to the tracked changes file. On comments where the change concerned a single word or phrase or multiple instances (e.g. all version numbers) we either give examples or do not mention line numbers where change occurred.

Reviewer #1 (Comments for the Authors (Required)):

It was pleasure for me to review the article. The authors adjusted the manuscript to the reviewers' suggestions and it was improved. No more suggestion.

Reviewer #2 (Comments for the Authors (Required)):

Thank you for revising your manuscript based on our comments. The manuscript is much improved from the previous version. The reanalyses have made the results much more convincing. Overall, it's a great paper and I think will be a very useful and welcome addition to the literature. I don't have any suggestions for reanalysis, but I do have some remaining comments on clarity and interpretation. My line numbers refer to the "Track Changes" document.

ABSTRACT

There are a few word/syntax choices that I would change:

"A dearth of" = "limited"

"Unbiased" = remove, and perhaps state "realised recombination"

"We find that recombination rates are correlated with..." would be less awkward

"We find no conclusive differences in..." recombination landscapes?

Sentences have been reworded based on suggestions from both reviewers.

SUMMARY

"Differences between" - "Differences in rate and landscape between"?

Added.

INTRODUCTION

In the Introduction, PRDM9 is introduced (Page 7) but I think in the updated introduction, it is now not explicitly mentioned that birds lack PRDM9 and have stable hotspots. It would be good to mention that somewhere as it is relevant for your study.

Added this information in the paragraph where PRDM9 is mentioned (lines 131-132)

Line 80: "it can be beneficial" might be better than "often"

Done

Line 81: delete "different"

Done

Lines 88-90: I didn't quite get this sentence - isn't genetic diversity also a consequence of biased-gene conversion?

Clarified to include gBGC and mutagenic effect of recombination (lines 88-90)

Line 121: "models"

Done

Line 136: species names should be in italics, would also precede those species "with some exceptions, such as [...]"

Corrected where applicable.

Line 170: "lagged behind" - this might be a little cheeky to the many groups working on recombination in birds :) maybe this could be toned down. I agree that it is rare to look at both contemporary and historical recombination, but actually this is rare for any species!

Toned down (lines 155-156)

Lines 185-195: When comparing map lengths, it's important to consider if they have used different mapping functions, especially with lower numbers of markers. This could be a reason why the map lengths could be quite different. FYI, a new study came out that also shows strong heterochiasmy in the hihi (very male biased!):

<https://www.nature.com/articles/s41437-024-00711-3>

We now cite the study where necessary. The importance of mapping functions is not stressed in our study but we toned down the highlighted part (lines 170-171).

RESULTS

I just wanted to comment that all the figures look great!

Lines 248 - briefly define a microchromosome?

Added (lines 229-230)

Line 281: "heterochiasmy among linkage groups at the chromosome scale"

Added.

Figure 1: I like the little stats box - could also add the number of COs?

Added.

Line 323: Not sure of the journal style, but I think it's good to put the version numbers of softwares in the results too (given that they come before the methods here)

Fixed where applicable.

Line 332 and throughout: I think you have to be careful about this statement about regions of low recombination in the linkage mapping. Low recombination regions are to be expected as there isn't a large number of crossovers in the dataset in the first place - perhaps a mean of 4COs per window per sex, meaning that loads of windows will have 0 COs just by chance. Looking at Figure S2, I see that a lot of these windows are adjacent, so you could incorporate this observation in your arguments about low recombination regions (I'm more convinced by long stretches, for example).

Indeed based on the linkage map we cannot reach any conclusions. However, the result is corroborated by the LD inference. Because these regions show reduced recombination we term them 'low recombination regions'. Where we claimed these regions to be "completely suppressed recombination" in the discussion we have toned it down (ex. Line 463) and we now note that its large stretches of reduced recombination.

Line 355: "leading to" should probably be "associated with"

Corrected.

DISCUSSION

Line 539: Just linking back to what I said above - these statements are a little risky without a bit more justification for "suppressed" recombination.

Here it was toned down (line 463), but the LD information supports the reduced recombination.

Lines 574 - 579: This is just a comment, but if the authors know - what do these dot chromosomes do during meiosis? Do they recombine as normal? Does anyone know?

Based on previous evidence they cluster together in a separate compartment where inter-chromosomal interactions are higher (Walters et al., 2021; Laia Marin-Gual et al., 2022).

Line 593: If I recall correctly, the great reed warbler had extreme telomeric recombination in males? It might be worth double checking this paper to make sure these were or were not different between the sexes.

Clarified now in lines 515-516.

Line 595: "dominated" isn't quite the right word - perhaps better to say that there was much higher recombination in females? Also going to flag again the hihi paper with male biased recombination :)

Fixed and added citation where relevant.

Line 600: "fine-scale information on sex differences"

Fixed.

Line 620: Why is haploid selection more relevant in plants? Maybe can say that there is no female haploid selection (if that's true) - I think this statement is slightly underdeveloped (although there is already a lot of detail in this section!)

This has been clarified and extended. Now in lines 539 - 544 we discuss why the potential for haploid selection is low and the predicted patterns not corroborated by findings.

Line 667: the sparrow paper will have a value for this chromosome as well (I don't know which chromosome it is!)

Added the sparrow value in lines 577-578.

Line 681: "were likely not inferred"

Fixed

Line 681 - 682: This sentence is a bit awkward, perhaps reword to say that the chromosomes that are less than 50cM were probably missing markers?

Reworded in lines 592-593

Line 690: yes - I think this sentence needs a reference for this statement though.

Added evidence from finches and other species (lines 607-608)

Lines 706 - 710: these are good points. I think these papers are more plant-centric so might be good to qualify that this may also hold for other species.

We added a few recent relevant references where this pattern is found (line 615).

Lines 712 - 713: maybe due to structural variants? I know that Ulrich Knief had a paper on inversions in zebrafishes that could be a good reference for that point.

Added evidence from finches and other species as in two comments above. Suggested work is mentioned where relevant.

Line 733: comma after "In fact"

Fixed

Line 741: "a few"

Fixed

Line 781: "inspired by"

Fixed

Line 786: "confounded"

Fixed

Line 794 - 795: Is this the case? I feel that it is ore that comparisons have been within PRDM9 or within non-PRDM9. That is kind of what the following sentences discuss.

Fixed the comparison sentence (line 691)

Line 807 - 808: as you don't develop this idea, perhaps the sentence can begin with a "yet".

Fixed

Line 879: no comma after "thus".

Fixed

METHODS

Caveating this section with - I am one of those people that read the methods before the results...!

Throughout - please provide ALL software version numbers, even if they were in the Results, and put all functions in italics. Also check the case (e.g. PLINK, Plink or plink?) of different softwares. There is a bit of a lack of consistency in these regards throughout the methods.

Corrected where relevant.

Lines 911 - 918: Put the sequencing coverage information here too

Added in line 742

Line 962 - 963: what does it mean to resolve first and second degree links?

Clarified in line 799

Line 976: "Mendelian"

Corrected.

Line 989: I haven't seen the use of the word "thrice" since I read Enid Blyton books, so maybe better to put "three times"

Fixed.

Line 993: "After confirming"

Fixed.

Line 1005: justification of 2cM?

We chose this value to remove outliers where linkage mapping might fail because of few flanking markers. It is not standardised anywhere so we went with a tradeoff of removing possible errors but not too many markers.

Line 1023: snowy owl (Latin name)?

Added.

Line 1032: Where is the Moran approximation implemented? E.g. software?

Clarified that it's a parameter of pyrho make table.

Line 1053: It has not been included at all how to calculate the intra-chromosomal allelic shuffling! It is in the Veller paper, also sparrows, Cathrine Brekke's recent papers, etc. - please put the equation and be clearer how this was done.

We have now included all the information we used to calculate it, including the equation in lines 882- 895.

DATA AVAILABILITY

Although a link to the code is signposted, there isn't an explicit data availability statement, please include one.

We include a data availability statement at the end of the manuscript with updated submission code and links | 928-935.

Reviewer #3 (Comments for the Authors (Required)):

I was pleased to see this manuscript again. The authors have improved it considerably, and it represents a useful and engaging addition to the scientific literature. It would benefit from a few more slight tweaks to better showcase its importance and novelty, and to clarify some methods, but is very close to ready.

General comments on substance:

The research presented here advances the scientific conversation in two directions: first, it improves the understanding of recombination hotspots in species that, like the majority of eukaryotes, lack PRDM9, and second, it expands the taxonomic breadth of (particularly birds) with recombination landscapes characterized, representing the first owl genome and only the third order of birds to be studied in this way.

Given that these are important and ongoing scientific conversations, the introduction could move faster to the importance of hotspots and recombination landscapes and condense the more general discussion of the importance of recombination and the methods used to infer it in favor of setting up the questions this work addresses.

While we have not restructured the introduction, we have highlighted the importance of studying recombination in birds by following more specific comments on the context (we highlight lack of knowledge in lines 155-156 and heterochiasmy in 181-184). We believe that by discussing the methods, their advantages and limitations we give some context on what each method can and has answered and how both can be used to improve conclusions. For example heterochiasmy is introduced under linkage mapping (l 109-110) and PRDM9 and hotspots under LD (l 125-130).

General comments on writing:

The writing is comprehensible but has some cumbersome moments. I've noted some in the specific comments below, but consider giving this a sharp copy edit. Also there are some typos and issues with the citation formatting, so make sure to clean those up.

Several sections, particularly in the discussion and conclusion, have a weak concluding sentence. For example, "Picking apart the magnitude and direction of these forces can help us understand what drives and what is driven by recombination rate variation in different species" is accurate, but doesn't add much insight - it just asserts that science can be done. In general, try to pep these up and perhaps mention some specific follow-up work (particularly the question of adaptive interspecific differences in recombination landscape and hotspot placement - that is interesting!).

We thank you for the improvement suggestions and have done our best to pep-up the concluding sentences (ex. 584-587, 624-626 644-648), especially in the conclusion (lines 717-720) . We have also added a dimension for future work on each topic. We note that on that specific paragraph we removed the concluding sentence and rephrased it to be more specific (lines 663-666)

Data availability:

The statement looks fine but the XXXX's need to be filled in, of course.

Filled in.

Specific comments:

Abstract:

The sentence featuring "albeit depending" is awkward.

"We find correlations of ..." - why not "gene content etc. correlated with"?

Sentences have been reworded based on suggestions from both reviewers.

Article summary:

Generally, a place for prose improvement.

Reworded for clarity.

Introduction:

I disagree that the inference of recombination rate is overlooked in non-model species. It's a major area of research right now. Perhaps it used to be neglected or is more challenging, but it isn't overlooked now.

We removed the statement.

The first three paragraphs could be condensed into a single paragraph to get to the novelty sooner. "Thus, to advance our understanding of recombination and distinguish it from other forces that shape the genome, we need to be able to quantify it and understand the sources of its variation in different species and under different evolutionary backgrounds" seems like a thesis statement that could steer that condensed paragraph.

We note that we had the paragraphs condensed and then expanded them in favour of discussing more implications of recombination in the last round of revisions. We have done our best to shorten this part in favour of the rest.

Does the (limited?) scientific understanding of non-PRDM9 mechanisms of recombination hotspots motivate this research?

This is partly true but because we shortened the PRDM9 related section of the introduction we have toned down this aspect.

It's really interesting that the patterns of heterochiasmy are all over the place in birds! That in itself is really interesting and motivating and could come earlier. It's a great reason to research bird recombination.

We agree and we have highlighted more the importance of studying heterochiasmy in birds in the introduction (lines 181-185).

Methods:

Comment more on these low-mappability regions. Are they annotated? If so, are they repetitive sequence? Where are they located? They're probably not important but add more justification for discarding them.

Added more information on this. Half of the masked regions are in unplaced scaffolds which did not make the linkage map (lines 777-779)

Why downsample using three scaffolds rather than pruned SNPs distributed across the genome? This is a strange choice.

When this filtering was performed we chose only 3 scaffolds to reduce the dataset since kinship should quickly converge to the genome-average value. We include here a difference of kinship matrices, one calculated using scaffolds 11,12 and 14 and the other calculated using genome-wide data. Values of differences range from -0.05 to 0.05 (see attached plot for full distribution). Since we used the genomic kinship to only resolve 1st and 2nd degree links (kinship of 0.125,0.25), the difference would not matter.

I may have missed it, but did any loci show evidence of transmission ratio distortion? I'd expect there to be some, and indeed one of the cited papers (Coop and Myers 2007) appears to be about how it relates to hotspots. It also relates to some of the theories described in Sardell and Kirkpatrick for landscape dimorphism. Anyway, if the filtering threw out distorted sites - I'm not sure how the bcftools Mendelian plugin works but if it was set to penalize divergence from HWE rather than just purge impossible sites - then an interesting dimension might have been lost.

We thank you for this suggestion and note that we have considered this interesting aspect. While hotspot transmission distortion is to be expected in PRDM9 species where the protein binds a strong motif and double strand breaks erode the motif in heterozygotes, we cannot think of an implication for hotspots in non-PRDM9 species. A very important aspect of transmission distortion which is indeed not tackled in our manuscript is GC-biased gene conversion and gene conversion in general which can be identified in family data. Because tracks of gene conversion are usually small and our coverage is often quite low (~10X) we did not attempt to quantify this here but aim to do so with a better dataset (higher coverage) in the future, along with mutation estimation for the species. Indeed as the manuscript and dataset are now this would require going back to raw data and adding another dimension. We believe the manuscript includes enough topics and further additions would make it hard to tackle all at the same time. Thus, these analyses as a whole are beyond the scope of this manuscript.

This is a hard thing because so many things affect recombination and can bias genotyping and inference, but how confident is it possible to be that these hotspots are not by chance?

We have tested the hotspot sharing qualitatively by using the sharing of the subsampled Swiss datasets as a measure of statistical inference noise. While hotspot sharing might not be caused by chance as sharing between subsets or different populations would not be expected, it can be caused by underlying features of the genome (repeats, low-mapping regions). Thanks to the reviewer's comments on the previous round of revisions we now prune our dataset more harshly and have shifted weight from any strong conclusions about hotspots in the population comparison. Thus, while pure chance can be ruled out, underlying confounding factors cannot. That's why we have exercised caution and have downscaled our conclusions based on these hotspots.

Maybe define the Gini coefficient briefly, since people may not be familiar with it outside the socioeconomic application? Especially since the relationship between Gini coefficient and length seems to be an effective way to describe these recombination patterns.

We have added a more formal definition in the results where we introduce the measure. (324-325)

Results:

Perhaps report slightly more about the putative centromeres in the text, especially since bird centromeres are poorly characterized and this is therefore a contribution to identifying those sequences? Since many of the chromosomes are acrocentric, just describe in a bit more detail what repeats were and were not observed in the low-recombination ends? Also, it is entirely fair that centromeres are really hard to assemble and annotate - even the human genome only got that done recently - but perhaps given that limitation the phrasing "near chromosome level" or the clarification "chromosome level, though not telomere-to-telomere" is merited.

The centromere annotation is described more in the results (271-275) and the discussion (546-549).

The not telomere-to-telomere aspect is now mentioned in Discussion (I 481).

It's confusing that in figure 2B the three chromosomes are not in fact at the same scale.

We assume the reviewer refers to the physical printed size of the x-axis not being identical among chromosomes? A bit of tweaking should have them very similar in the new version. If the comment is referring to the x-axis being in Mb instead of scaled to length (from 0 to 1) we preferred to portray it like this to show the effect of the length on the distribution of recombination rates.

Discussion:

Slightly reordering the sentences about the regular vs. dot chromosomes would be much clearer. "The karyotype of the barn owl contains 45 autosomal pairs, including a large Z chromosome, small degraded W chromosome, and six 'dot' microchromosomes." Then we kind of expect the 39 that were actually assembled. Also, how much evidence is there that the dot chromosomes are biologically relevant? Could the discussion of them be compressed?

We have re-arranged these sentences as suggested. (I 475-483)

The dot chromosomes following the investigation of Huang et al., 2023 in the chicken are found to be gene-rich, GC-rich elements, probably enriched in housekeeping genes. While in the grand scheme of things they might not be the most relevant genomic entities as they represent a limited number of genes, we choose to maintain this part of the discussion as it might be useful for future studies.

Some of the context about recombination landscapes, how little is known about heterochiasmy in birds, and the implications of sex differences in landscape could perhaps

be set up in the introduction. Similarly, the section describing inter- and intraspecific variation in recombination landscape is really interesting and positions this in the context of broader comparative work, so a hint of it could be used to motivate the study.

We have highlighted the importance of studying heterochiasmy in birds in the introduction (lines 180-185).

November 1, 2024

RE: GENETICS-2024-307579

Mr. Alexandros Topaloudis
Universite de Lausanne
Department of Ecology and Evolution
Quartier UNIL-Sorge Bâtiment Biophore
Lausanne, N/A CH-1015
Switzerland

Dear Dr. Topaloudis:

Congratulations! We are delighted to inform you that your manuscript entitled "The recombination landscape of the barn owl, from families to populations" is acceptable for publication in GENETICS. Many thanks for submitting your research to the journal.

To Proceed to Production:

1. Format your article according to GENETICS style, as discussed at <https://academic.oup.com/genetics/pages/general-instructions>, and upload your final files at <https://genetics.msubmit.net>.
2. Your manuscript will be published as-is (unedited-as submitted, reviewed, and accepted) at the GENETICS website as an Advanced Access article and deposited into PubMed shortly after receipt of source files and the completed license to publish. Please notify sourcefiles@thegsajournals.org if you do not wish to publish your article via Advanced Access.
3. We invite you to submit an original color figure related to your paper for consideration as cover art. Please email your submission to the editorial office or upload it with your final files. You can submit a small-sized image for evaluation, and if selected, the final image must be a TIFF file 2513px wide by 3263px high (8.375 by 10.875 inches; resolution of 600ppi). Please avoid graphs and small type.

If you have any questions or encounter any problems while uploading your accepted manuscript files, please email the editorial office at sourcefiles@thegsajournals.org.

Sincerely,

Stephen Wright
Associate Editor
GENETICS

Approved by:
David Begun
Senior Editor
GENETICS

note: Please add jnls.author.support@oup.com and genetics.oup@kwgglobal.com (or the domains @oup.com and @kwgglobal.com) to your email program's "safe senders" list. You will be contacted by both at various points during the production process.

Review comments (if applicable):